# DUAL PERSPECTIVES ON NON-CONTRASTIVE SELF-SUPERVISED LEARNING

**Jean Ponce**
Ecole normale supérieure/PSL
New York University
`jean.ponce@ens.fr`

**Basile Terver**
Meta FAIR
INRIA Paris
`basileterv@meta.com`

**Martial Hebert**
Carnegie-Mellon University
`martial.hebert@cs.cmu.edu`

**Michael Arbel**
Univ. Grenoble Alpes, Inria
CNRS, Grenoble INP, LJK
`michael.arbel@inria.fr`

## ABSTRACT

The *stop gradient* and *exponential moving average* iterative procedures are commonly used in non-contrastive approaches to self-supervised learning to avoid representation collapse, with excellent performance in downstream applications in practice. This presentation investigates these procedures from the dual viewpoints of optimization and dynamical systems. We show that, in general, although they *do not* optimize the original objective, or *any* other smooth function, they *do* avoid collapse. Following Tian et al. (2021), but without any of the extra assumptions used in their proofs, we then show using a dynamical system perspective that, in the linear case, minimizing the original objective function without the use of a stop gradient or exponential moving average *always* leads to collapse. Conversely, we characterize explicitly the equilibria of the dynamical systems associated with these two procedures in this linear setting as algebraic varieties in their parameter space, and show that they are, in general, *asymptotically stable*. Our theoretical findings are illustrated by empirical experiments with real and synthetic data.

## 1  INTRODUCTION

*Self-supervised learning* (or *SSL*) is an approach to representation learning that exploits the internal consistency of training data *without* requiring expensive annotations. It has proven to be an effective alternative to traditional supervised technology, for applications in natural language processing (Mikolov et al., 2013; Vaswani et al., 2017), image analysis (Chen et al., 2020; Grill et al., 2020; Caron et al., 2021; Chen and He, 2021; Radford et al., 2021; Zbontar et al., 2021; Bardes et al., 2022) and video understanding (Bardes et al., 2024). Early SSL approaches, e.g., (Mikolov et al., 2013; Chen et al., 2020; He et al., 2020; Radford et al., 2021), were *contrastive*: models are learned from training pairs that can be either *negative*, when one data point is representative of the target population and the other one is not, or *positive*, when both data points are representative of the target. The training consists in pushing negative pairs apart while pulling positive ones together. *Non-contrastive* approaches to SSL have emerged as a powerful alternative often outperforming contrastive ones empirically (Grill et al., 2020; Chen and He, 2021; Bardes et al., 2024; Assran et al., 2023; Bardes et al., 2022; Caron et al., 2021). These techniques compute representations of different *views* of the same data and learn to predict one from another, thus avoiding the need for mining negative data. They are, however, susceptible to *representational collapse* where a constant embedding is learned (LeCun, 2022).

Preventing representation collapse in non-contrastive approaches has thus become a key focus in SSL, leading to two principal strategies: *feature decorrelation* and *enforcing asymmetry between the two views*. The first strategy addresses representational collapse by explicitly enforcing decorrelation among the learned features. For instance, Bardes et al. (2022) introduce a regularizer $\Omega$ designed to avoid collapse by keeping the variance of the codes of the two views of samples above

a fixed threshold while encouraging the codes associated with the same sample to be similar. More recently, Sansone et al. (2025) proposed an auxiliary classification task with randomly assigned labels, providing theoretical guarantees against collapse. Despite their conceptual simplicity, feature-decorrelation methods are often empirically outperformed by techniques that introduce asymmetries between the views during training to avoid collapse. Specifically, these rely on a teacher/student architecture, where the student computes a *source* view as the composition of encoder and predictor networks and aims to predict a *target* view obtained using the teacher network. The latter is either a frozen copy (through a *stop gradient* operation or *SG*) or a delayed version (through an *exponential moving average* or *EMA*) of the student encoder (Oquab et al., 2024; Bardes et al., 2024; Assran et al., 2023; Grill et al., 2020; Chen and He, 2021). SG and EMA have shown strong empirical performance and remain standard components of state-of-the-art SSL models (Assran et al., 2023; Oquab et al., 2024).

**Problem statement.** Despite the empirical success of SG and EMA, there is no obvious link between these methods and the optimization of a well-defined objective function. This motivates gaining a theoretical understanding of their behavior, including: (a) Do SG and EMA solve an optimization problem and, if they do, which one? (b) Do they converge and, when and if they do, are they guaranteed to avoid collapse? (c) Seen as dynamical systems, are their stationary points, if any, stable, so there is no risk of drifting from them to some trivial solution? These are the problems we address in the rest of this presentation, from dual perspectives: an *optimization perspective* for (a) and (b), and a *dynamical system* one for (c), following the work of (Littwin et al., 2024; Tian et al., 2021; Wang et al., 2021) in the *linear* case.

**Main contributions.**

(1) We prove with Proposition 3.1 that, in general,[1] neither the SG algorithm nor its EMA counterpart minimizes the objective they are derived from and that, if they converge, they both avoid collapse (Figure 1).

(2) In the case where the loss is the squared Euclidean distance, we then prove with Proposition 3.2 the conjecture given in (Grill et al., 2020) that the SG and EMA algorithms do not optimize *any* well defined function.

(3) We confirm (1) empirically (Section 3.3) on an action classification task from video data, further finding that the SG and EMA algorithms do not appear to converge, although their downstream performance increases momentarily in training.

Following Littwin et al. (2024); Tian et al. (2021); Wang et al. (2021), we then switch to a dynamical system perspective in the case where the encoder and predictor are both linear operators.

(4) We characterize in Proposition 4.5 and Corollary 4.6 the equilibria of the dynamical systems associated with the SG and EMA algorithms as a finite set of algebraic varieties.

(5) We prove with Proposition 4.10 that these equilibria are, in general, *asymptotically stable* (Arnol'd, 1992). In particular, when started close to them, the two procedures are guaranteed to converge there and stay there.

(6) We run some simulations in the simplified setting where the input space is scalar ($m = 1$) and show that the two algorithms converge in general in these experiments, although possibly to trivial minima.

All proofs are relegated to the appendix for conciseness. In the linear case, these proofs leverage the equations' structure (Petersen and Pedersen, 2012) to avoid cumbersome tensor manipulations. On a minor note, this notably allows us to rederive, for completeness, several results about the dynamics of the SG and EMA algorithms (Lemmas 4.1 to 4.4) already known from (Tian et al., 2021), but without assumptions from their original proofs, such as that the two views of the data be drawn from the same distribution conditioned on the data, or that the eigenvalues of certain PSD matrices be bounded away from zero, whose validity is difficult to guarantee in practice.

---

[1] Whenever we state that some property holds *in general*, this means that, although they may not hold for *certain* data satisfying specific equations, they do hold, in practice, for *all* generic data, in the standard mathematical sense, following the common notion of genericity in dynamical systems and algebraic geometry, e.g., Hirsch et al. (2013).

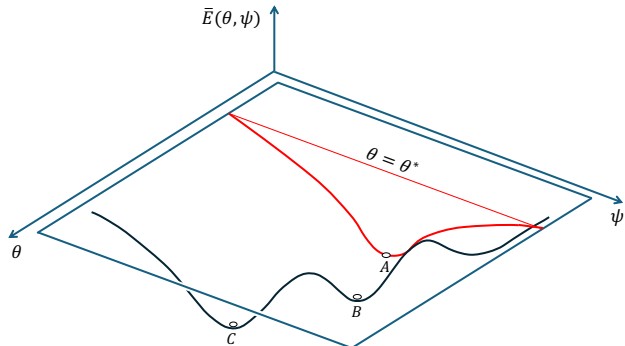

Figure 1: A (toy) illustration of the optimization landscape for the objective function $\bar{E}(\theta, \psi)$ of Eq. (1). Here $C$ is the global minimum of $\bar{E}(\theta, \psi)$ (shown as negative instead of zero for readability) associated with a collapse of the training process; $B$ is a nontrivial local minimum one may reach using an appropriate regularization to avoid collapse; and $A$ is a limit point of the stop gradient (SG) training procedure associated with parameters $\theta^*$ and $\psi^*$ at convergence. In general, it is not a minimum of $\bar{E}$ and thus does not correspond to a collapse of the training process, but it is a minimum with respect to $\psi$ of $\bar{E}(\theta^*, \psi)$. See text for details.

## 1.1 RELATED WORK

Several works have explored why SSL methods learn effective representations, with feature-decorrelation methods being specifically investigated in (Balestriero and LeCun, 2022; Weng et al., 2022; Ziyin et al., 2023; Jing et al., 2022). Balestriero and LeCun (2022) attempt to unify VICReg (Bardes et al., 2022), SimCLR (Chen et al., 2020) and Barlow Twins (Zbontar et al., 2021) in a single framework using global and local spectral embedding methods. Weng et al. (2022) investigate how whitening-based losses avoid collapse. Ziyin et al. (2023) develop an analytically tractable theory of SSL loss landscapes for both contrastive and feature-decorrelation methods, analyzing factors that affect SSL robustness to data imbalance.

Experimental studies of asymmetry-based methods (Tao et al., 2021; Zhang et al., 2022; Liu et al., 2022) have investigated their properties. Notably, UniGrad (Tao et al., 2021) systematically compares contrastive and non-contrastive SSL approaches, concluding that a momentum encoder is key to improved performance. Zhang et al. (2022) use gradient analysis of l2-normalized representations to study why SimSiam avoids collapse. Liu et al. (2022) experimentally show that asymmetry-based methods like BYOL and SimSiam implicitly enforce feature decorrelation, linking them to methods such as Barlow Twins and VICReg.

Theoretical investigations of asymmetry-based methods have been proposed, notably by Tian et al. (2021), showing that, under isotropic data and specific optimization trajectory assumptions, the predictor and stop-gradient are essential to prevent collapse in linear BYOL and SimSiam. This study has inspired algorithms where the predictor is an explicit function of the encoder, with proven collapse-avoiding properties (Wang et al., 2021; Jing et al., 2022; Halvagal et al., 2023; Tang et al., 2023). Wen and Li (2022) investigate the encoder's role in avoiding collapse using a two-layer neural network, albeit under a simplified data-generating process. Finally, Littwin et al. (2024) study the effect of depth on learned representations in deep linear models, showing that Joint-Embedding Predictive Architectures (or JEPA) models prioritize "influential" features (features which are most informative in prediction), with derivations relying on diagonal covariance matrices and orthogonal initialization.

## 2 PROBLEM SETTING

Given a parametric encoder $f_\theta : \mathbb{R}^m \to \mathbb{R}^n$ and a parametric predictor $g_\psi : \mathbb{R}^n \to \mathbb{R}^n$ with parameters $\theta$ in $\mathbb{R}^p$ and $\psi$ in $\mathbb{R}^q$, it is possible (LeCun, 2022; Bardes et al., 2022) to learn $\theta$ and $\psi$ from data embedded in $\mathbb{R}^m$ without outside supervision by minimizing with respect to these parameters the objective function

$$\bar{E}(\theta, \psi) = \mathbb{E}_{x,y} E(\theta, \psi, x, y), \text{ where } E(\theta, \psi, x, y) = l[g_\psi \circ f_\theta(x), f_\theta(y)] + \Omega(\theta, \psi). \quad (1)$$

Here, $x$ and $y$ are different *views* of some data point (e.g., different crops of the same image), $\mathbb{E}_{x,y} E$ is the mean of the function $E$ over the (unknown) distribution of the views conditioned on the corresponding data, approximated in practice by a mean over a finite number of data samples,

$l : \mathbb{R}^n \times \mathbb{R}^n \to \mathbb{R}^+$ is some loss and $\Omega : \mathbb{R}^p \times \mathbb{R}^q \to \mathbb{R}^+$ is some regularizer. The corresponding architecture is that of a Siamese network (Bromley et al., 1994) whose branches correspond to the encoders of the two views compared, with shared parameters, while the predictor sits on top of the first branch. In this setting, $\bar{E}(\theta, \psi)$ can be minimized with respect to these parameters by using, for example, stochastic gradient descent. With proper learning rates, the training procedure will converge to some critical point of $\bar{E}$ where both gradients are zero (Figure 1). A difficulty, however, is how to prevent it from *collapsing* by converging to the degenerate zero global minimum corresponding to $f_\theta$ being a constant and $g_\psi$ being the identity, or $f_\theta$ being zero and $g_\psi$ being a function such that $g_\psi(0) = 0$. To address this difficulty, *BYOL* (Grill et al., 2020) and *SimSiam* (Chen and He, 2021) propose to use *exponential moving average* and *stop gradient* training procedures, as defined in the rest of this section, as alternative minimization procedures. To properly define these procedures, let us introduce an objective function with an additional argument $\xi$ in $\mathbb{R}^p$:

$$\bar{F}(\theta, \psi, \xi) = \mathbb{E}_{x,y} F(\theta, \psi, \xi, x, y), \text{ where } F(\theta, \psi, \xi, x, y) = l[g_\psi \circ f_\theta(x), f_\xi(y)] + \Omega(\theta, \psi), \quad (2)$$

and consider instead the *exponential moving average* (*EMA*) procedure (Grill et al., 2020).

---

**EMA algorithm:** Initialize $\theta_0$, $\psi_0$ and $\xi_0$ to some values and $t$ to 1, then repeat until convergence or you run out of patience:

    (a) $\theta_t \leftarrow \theta_{t-1} - \mu_t \nabla_\theta \bar{F}(\theta_{t-1}, \psi_{t-1}, \xi_{t-1})$;

    (b) $\psi_t \leftarrow \psi_{t-1} - \nu_t \nabla_\psi \bar{F}(\theta_{t-1}, \psi_{t-1}, \xi_{t-1})$;

    (c) $\xi_t \leftarrow \alpha_t \xi_{t-1} + (1 - \alpha_t)\theta_t$;

    (d) $t \leftarrow t + 1$.

---

This is the procedure used to train BYOL in (Grill et al., 2020) and V-JEPA in (Bardes et al., 2024)[2]. The corresponding architecture is no longer a true Siamese network because the encoders in its two branches have different parameters, so $f_\theta$ and $f_\xi$ are sometimes respectively called the *online* (or *student*) and *target* (or *teacher*) networks. An alternative is to consider the *stop gradient* procedure (Chen and He, 2021) used to train SimSiam, which is a true Siamese architecture with identical encoders in its two branches but uses $\nabla_\theta \bar{F}$ as a proxy for $\nabla_\theta \bar{E}$ when updating $\theta$.

---

**SG algorithm:** Initialize $\theta_0$ and $\psi_0$ to some values and $t$ to 1, then repeat until convergence or you run out of patience:

    (a) $\theta_t \leftarrow \theta_{t-1} - \mu_t \nabla_\theta \bar{F}(\theta_{t-1}, \psi_{t-1}, \theta_{t-1})$;

    (b) $\psi_t \leftarrow \psi_{t-1} - \nu_t \nabla_\psi \bar{F}(\theta_{t-1}, \psi_{t-1}, \theta_{t-1})$;

    (c) $t \leftarrow t + 1$.

---

Note that $\bar{E}(\theta, \psi) = \bar{F}(\theta, \psi, \theta)$ for any values of $\theta$ and $\psi$, but it is a priori unclear whether any limit point of the SG procedure is related to the critical points of $\bar{E}$. Note also that the EMA algorithm reduces to the SG one when $\alpha_t = 0$ and $\xi_0 = \theta_0$. Indeed, Chen and He (2021) refer to SimSiam as "BYOL *without* the momentum encoder", but in practice $\alpha_t$ is taken very close to 1, e.g., $\alpha_t$ varies between 0.996 and 1 in BYOL (Grill et al., 2020).

## 3 AN OPTIMIZATION PERSPECTIVE

*We assume from now on for simplicity that $\alpha_t$ is a constant with $\alpha_t \neq 1$ or converges to such a constant as $t$ tends to infinity.* Under this assumption, if and when the EMA and SG procedures converge, we have by continuity $\theta = \xi$ at a limit point because of step (c), with the same gradient values in $\theta$ and $\psi$ as in SG. The dynamics may be different, and lead to different limit points, but these will obey the *same* zero gradient conditions at the equilibria of the corresponding dynamical systems.

---

[2]BYOL (Grill et al., 2020) assumes that what we call an encoder is a proper encoder followed by a projection operator and that the loss acts on normalized versions of its inputs. Both BYOL and SimSiam (Chen and He, 2021) make the loss symmetric by having each view predict the other. This is subsumed by the framework presented here and does not change the conclusions of our analysis

## 3.1 EMA AND SG OBJECTIVES

The *motivation* for the EMA algorithm in BYOL (Grill et al., 2020, Sec. 3.1, Eq. (2)) is to minimize over $\theta$, $\psi$ and $\xi$ the mean over pairs of views $x$ and $y$ of the data the squared Euclidean distance between the *prediction* $g_\psi \circ f_\theta(x)$ obtained from the representation of $x$ and the *target* encoding $f_\xi(y)$ of $y$, where $\xi$ is the moving average defined recursively by step (c) of EMA. Note that, although $\xi_t$ is well defined at each $t$, an explicit definition of $\xi$ is missing, except in the limit case if and when EMA converges. It is thus *a priori* unclear whether its iterations minimize a well-defined function. Similarly, the motivation for the SG training procedure in (Chen and He, 2021, Sec. 3, Eq. (1)) is to minimize the mean squared Euclidean distance between the prediction $g_\psi \circ f_\theta(x)$ obtained from $x$ and the target $f_\theta(y)$, using $\nabla_\theta \bar{F}$ as a *proxy* for the true gradient $\nabla_\theta \bar{E}$ of the corresponding objective function. It is therefore again *a priori* unclear whether it minimizes a well-defined function.

## 3.2 SG AND EMA ALGORITHMS VS OPTIMIZATION PROBLEMS

It is important for fairness to clearly state that none of the papers we are aware of that use the SG or EMA procedure claims to actually minimize such functions in practice. In fact, the BYOL authors write (Grill et al., 2020): "More generally, we hypothesize that there is no loss $L_{\theta,\xi}$ such that BYOL's dynamics is a gradient descent on $L$ jointly over $\theta, \xi$." One of our main contributions is actually to prove this conjecture (see Proposition 3.2 below). Let us first state a simple but important result (see the appendix for its proof).

**Proposition 3.1.** *The SG and EMA algorithms do not, in general, minimize the original objective function $\bar{E}$ of Eq. (1). If and when they converge, the corresponding solution is, in general, not a degenerate one corresponding to a zero global minimum of that function.*

A harder question to answer is whether these algorithms optimize *any* objective function. *Let us assume from now on* for simplicity that $l$ is the (half) squared Euclidean distance and $\Omega(\theta, \psi) = \lambda(\|\theta\|^2 + \|\psi\|^2)/2$. With this choice, we have $\nabla_u l(u,v) = u - v$, $\nabla_\theta \Omega = \theta$, $\nabla_\psi \Omega = \psi$ and

$$\begin{cases} \nabla_\theta F(\theta, \psi, \theta, x, y) = J_\theta u(\theta, \psi, x)^T [u(\theta, \psi, x) - v(\theta, y)] + \lambda\theta, \\ \nabla_\psi F(\theta, \psi, \theta, x, y) = J_\psi u(\theta, \psi, x)^T [u(\theta, \psi, x) - v(\theta, y)] + \lambda\psi, \end{cases} \tag{3}$$

where $u(\theta, \psi, x) = g_\psi \circ f_\theta(x)$ and $v(\theta, y) = f_\xi(y)$. We wish to understand whether the vector fields $\mathbb{E}_{x,y}[\nabla_\theta F(\theta, \psi, x, y)]$ and $\mathbb{E}_{x,y}[\nabla_\psi F(\theta, \psi, x, y)]$ are the gradients of some well defined scalar function which the SG algorithm (and its EMA counterpart) would presumably minimize. The answer is negative:

**Proposition 3.2.** *Under the (mild) assumption that $f_\theta$ and $g_\psi$ are smooth neural networks whose last layer is linear and that they are not identically 0, i.e. there exists $\theta_0, \psi_0, x_0$ and $z_0$ so that $f_{\theta_0}(x_0) \neq 0$ and $g_{\psi_0}(z_0) \neq 0$, then the vector fields $\mathbb{E}_{x,y}[\nabla_\theta F(\theta, \psi, x, y)]$ and $\mathbb{E}_{x,y}[\nabla_\psi F(\theta, \psi, x, y)]$ are not, in general, the gradient fields of any smooth function.*

*Proof sketch.* The full proof, given in the appendix, follows from Eq. (3) and Schwarz's integrability theorem, according to which, a *necessary* condition for these vector fields to be the gradient field of a smooth scalar function is that their second-order cross derivatives be the transposes of each other. By expressing these cross derivatives we find that their difference (taking into account the transposition operation) does not vanish in a generic sense. To establish genericity, we show that there exists arbitrarily small perturbations to the data distribution for which the corresponding cross derivative difference cannot be identically zero. $\square$

## 3.3 EXPERIMENTS WITH REAL DATA

We investigate in this section three fundamental questions about the SG and EMA algorithms: (1) Although they do not, in theory, minimize the original objective $\bar{E}(\theta, \psi)$, do they minimize it in practice? (2) Do they converge, and in particular do $\theta_t - \theta_{t-1}$ and $\psi_t - \psi_{t-1}$ both tend toward zero as the number of training steps increases? (3) Does the classification accuracy increase with training time? We address these three questions in a realistic setting with experiments on a video classification task using the code of Bardes et al. (2024), on the Kinetics710 and SSv2 benchmarks (Goyal et al., 2017; Smaira et al., 2020).

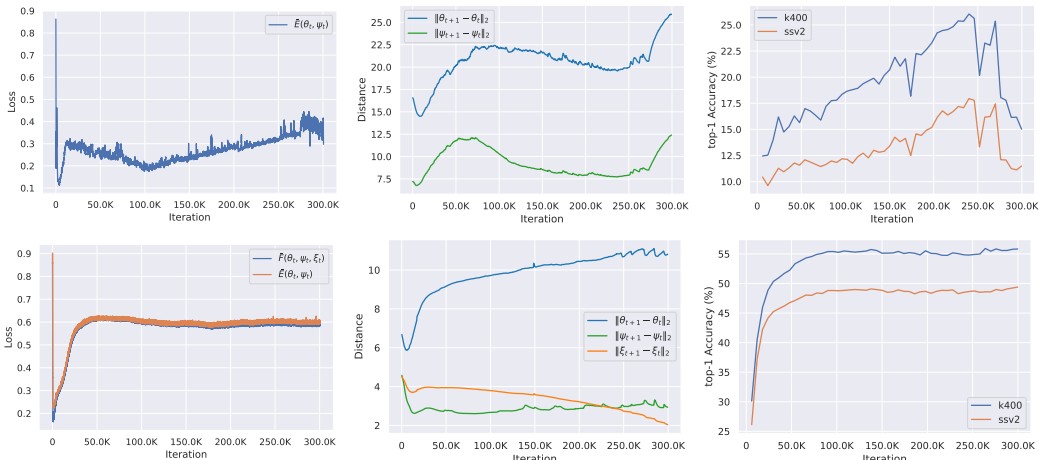

Figure 2: Evolution over 300,000 iterations of (left) the $\bar{E}$ objective (plus $\bar{F}$ for EMA) without the squared Euclidean norm regularizer, (middle) the norms of the parameter increments $\theta_t - \theta_{t-1}$, $\psi_t - \psi_{t-1}$ and $\xi_t - \xi_{t-1}$, and (right) the classification accuracy on the K400 and SSv2 benchmarks for SG (top) and EMA (bottom). All curves here are smoothed using a moving average, and the squared Euclidean norm regularizer is ignored to emphasize the main loss.

**Experimental setting.** To provide a realistic setting, we address the challenging problem of self-supervised learning for action classification in videos, a task for which VICReg (Bardes et al., 2022) and regularization-based methods such as SimCLR (Chen et al., 2020) and Barlow Twins (Zbontar et al., 2021) have not proven successful *so far*. Indeed, the state-of-the-art V-JEPA model of Bardes et al. (2024) uses the EMA algorithm for training, and we use its code, kindly provided by its authors, in our experiments, with an $\ell_1$ loss for $l$, a ViT-S encoder and a ViT-T predictor (Dosovitskiy et al., 2021). We use the unions of the Kinetics710 and SSv2 training datasets (Goyal et al., 2017; Smaira et al., 2020) for learning the representation. We also learn in a supervised manner an attentive pooling classifier as in (Bardes et al., 2024), with the train splits of each dataset. We report the top-1 accuracy on the classification task for each dataset on its validation splits.

We slightly modify the training setup compared to (Bardes et al., 2024) to make it simpler and faster. We run the SG and EMA algorithms on both datasets in our experiments, with 1000 training epochs (300 000 iterations), instead of 300 epochs in V-JEPA. We also restrict the train set to Kinetics710 and SSv2, discarding Howto100M Miech et al. (2019). We use a ViT-Base encoder instead of a ViT-L encoder and reduce the batch size to 1024. Instead of the cosine annealing with warmup schedulers, we fix the learning rate to 0.0001 and the weight decay to 0.1. Following (Bardes et al., 2024), the variable $\alpha_t$ increases from 0.998 to 0.9996 across iterations. As expected, with these simplifications, the performance of the trained models on the downstream K400 and on SSv2 video classification tasks (Figure 2, bottom) is lower than the accuracies of 72.9 and 67.4 reported in (Bardes et al., 2024) for the V-JEPA ViT-L model trained on K710 and SSv2 for a total of 900K samples seen during training. Yet we believe that this simple setting is sufficient to run realistic experiments and give preliminary yet meaningful answers to the questions addressed in this section.

**Results and conclusions.** Figure 2 (top, left) shows the evolution of $\bar{E}(\theta_t, \psi_t)$ for the SG procedure as a function of the number of iterations. As expected, the algorithm does not minimize the function, its smallest value being reached early in the iterations. Figure 2 (bottom, left) shows the evolution of $\bar{F}(\theta_t, \psi_t, \xi_t)$ and $\bar{E}(\theta_t, \psi_t)$ for the EMA procedure, and similar conclusions can be reached. As shown in Figure 2 (top, middle), $\theta$ and $\psi$ *do not* converge in our experiments either since the norms of $\theta_t - \theta_{t-1}$ and $\psi_t - \psi_{t-1}$ do not tend to zero. As shown in Figure 2 (bottom, middle), a similar conclusion can be reached for $\theta$, $\psi$ and $\xi$ and the EMA algorithm. Finally, classification accuracy *does* increase initially for both the SG and EMA algorithms before decreasing late in the process for SG, and reaching a plateau for EMA (Figure 2, right), confirming the commonly accepted fact that they learn something, although it is still unclear exactly what they learn. In this *particular* setting, EMA gives better classification results than SG. *General* conclusions about other downstream tasks should not be drawn from this of course.

## 4 A DYNAMICAL SYSTEM PERSPECTIVE IN THE LINEAR CASE

Let us now consider, following (Littwin et al., 2024; Tian et al., 2021; Wang et al., 2021), the linear case where $f_\theta(x) = Ax$, $f_\xi(y) = Cy$, and $g_\psi(z) = Bz$, where $x \in \mathbb{R}^m$, $A$ and $C$ are $n \times m$ matrices, with $n > m$, $z \in \mathbb{R}^n$, $B$ is an $n \times n$ matrix, and the vectors $\theta$, $\xi$ and $\psi$ store row after row the coefficients of $A$, $C$ and $B$. We will from now on write $[xx^T]$ for $\mathbb{E}_x[xx^T]$ and $[yx^T]$ for $\mathbb{E}_{x,y}[yx^T]$. Recall that we consider, here, the (half) squared Euclidean distance for $l$ and $\Omega(\theta, \psi) = \lambda(\|\theta\|^2 + \|\psi\|^2)/2$.

### 4.1 THE SG AND EMA ALGORITHMS AS DYNAMICAL SYSTEMS

An algorithm such as the SG and EMA procedures can also be viewed as a discrete dynamical system. In this case, the dynamics are driven by the gradients of the function $F$ with respect to the parameters being optimized, *i.e.*, the matrices $A$, $B$ and $C$. Note: *We assume from now on that $\alpha_t$ is a constant $\alpha \neq 1$.* For completeness, we rederive in the rest of this section several results about the dynamics of the SG and EMA algorithms (Lemmas 4.1 to 4.4) already known from (Tian et al., 2021). As noted earlier, however, our proofs, given in appendix, do not rely on the assumptions required by the original ones, for example that the two views of the data be drawn from the same distribution conditioned on the data, or that the eigenvalues of certain PSD matrices be bounded away from zero, whose validity is difficult to guarantee in practice.

**Lemma 4.1.** *The discrete dynamics of the EMA algorithm in the linear case are given by*

$$\begin{cases} A_t = A_{t-1} - \mu_t(B_{t-1}^T R(A_{t-1}, B_{t-1}, C_{t-1}) + \lambda A_{t-1}), \\ B_t = B_{t-1} - \nu_t(R(A_{t-1}, B_{t-1}, C_{t-1})A_{t-1}^T + \lambda B_{t-1}), \\ C_t = \alpha C_{t-1} + (1 - \alpha)A_t, \end{cases} \quad (4)$$

*where $R(A, B, C) \stackrel{\text{def}}{=} BA[xx^T] - C[yx^T]$.*

The dynamics for the SG algorithm are obtained by taking $C = A$ in Eq. (4) and not using the $C$ update. The stationary points of SG must satisfy the $p + q$ equations in $p + q$ unknowns defined by $\bar{P} = 0$ and $\bar{Q} = 0$, with $p = nm$ and $q = n^2$, whose solutions include $A = 0$ and $B = 0$. *In particular, unlike the nonlinear case, it is* a priori *possible in the linear setting for a limit point of the SG or EMA algorithm to be the degenerate global minimum associated with $A = 0$ and $B = 0$.*[3] The following lemma follows easily from Eq. (4) and is reminiscent of prior results on gradient descent for deep linear networks (Baldi and Hornik, 1989).

**Lemma 4.2.** *When $\lambda > 0$, the limit points of the SG algorithm, if they exist, satisfy $B^T B = AA^T$.*

We switch now to a continuous dynamical system perspective, following (Littwin et al., 2024; Tian et al., 2021; Wang et al., 2021), to simplify the analysis. The continuous version of Eq. (4) is

$$\begin{cases} \dot{A} = -(B^T R(A, B, C) + \lambda A), \\ \dot{B} = -(R(A, B, C)A^T + \lambda B), \\ \dot{C} = (1 - \alpha)(A - C). \end{cases} \quad (5)$$

where $R(A, B, C)$ is the same as in Lemma 4.1. At a limit point, we have in addition $\dot{A} = 0$, $\dot{B} = 0$ and multiplying the first equation on the right by $A^T$ and the second one on the left by $B^T$, then subtracting the results yields $B^T B = AA^T$. Given Eq. (5) we can now rederive two results from (Tian et al., 2021).

**Lemma 4.3.** *Given the dynamical system associated with the gradient flow for the original objective function of Eq. (1), the matrix $A$ always converges to zero.*

This is Theorem 2 in Tian et al. (2021), but without the assumptions made in the original proof. This result implies that there is no other limit point than the global trivial minima where $A = 0$ and $B$ can assume any value. Tian et al. (2021) (and (Littwin et al., 2024; Wang et al., 2021)) interpret Eq. (5) as defining a *gradient flow*. Note that it might be more appropriate to see these equations as defining

---

[3]The degenerate solution where $f_\theta$ is a nonzero constant and $g_\psi$ is the identity does not occur in the linear case since $Ax$ constant implies $A = 0$.

a general *flow* instead, that is, a first-order ordinary equation whose integral curves are the output of the algorithm since we have shown that their corresponding vector fields are not the gradients of a well defined function by Proposition 3.2. We can now state in our setting Theorem 1 from (Tian et al., 2021) as follows.

**Lemma 4.4.** *Under the SG or EMA dynamics, the difference of the two matrices $B^T B$ and $AA^T$ tends to zero as $t$ tends to infinity.*

## 4.2 Characterizing the equilibria of the SG and EMA algorithms

**Proposition 4.5.** *Assuming that the matrix $A$ has maximal rank $m < n$ at equilibria of the SG algorithm, these equilibria are the $(A, B)$ pairs such that*

$$([xx^T]S + \lambda Id)(S[xx^T] + \lambda Id) = [xy^T]S[yx^T], \text{ where } S = A^T A. \tag{6}$$

*and*

$$B = A[yx^T]A^T W^{-1} \text{ where } W = A[xx^T]A^T + \lambda Id. \tag{7}$$

*Assuming again that the matrix $A$ has maximal rank $m < n$ at the equilibria of the EMA algorithm, these equilibria are the $(A, B, C)$ triples where $A$ verifies Eq. (6), $B$ is given by Eq. (7) and $C = A$. In both cases, the $(A, B)$ pairs associated with equilibria also verify $B^T B = AA^T$.*

Equation (6) is a system of $m(m+1)/2$ quadratic equations in the $m(m+1)/2$ independent entries of the symmetric matrix $S$, which is positive definite since $A$ is assumed to have maximal rank. Such a system admits in general at most $2^{m(m+1)/2}$ solutions, and we obtain the following result.

**Corollary 4.6.** *Let $K$ denote the number of distinct real solutions $S_k$ $(k = 1, \ldots, K)$ of Eq. (6) such that $S_k$ is positive definite and, for $k = 1, \ldots, K$, let $\sqrt{S_k}$ denote the unique positive definite square root of $S_k$. The equilibria of the SG algorithm can be decomposed into $K$ sub-varieties of $\mathbb{R}^{n \times m} \times \mathbb{R}^{m \times m}$ formed by pairs $(A, B)$ such that $A$ belongs to*

$$\mathcal{A}_k = \{A \in \mathbb{R}^{n \times m}, \ A^T A = S_k\} = \{U\sqrt{S_k}, \text{ where } U \in \mathbb{R}^{n \times m} \text{ and } U^T U = Id\}, \tag{8}$$

*and $B$ satisfies Eq. (7). The equilibria $(A, B, C)$ of the EMA algorithm can be characterized in a similar way to $A$ in $\mathcal{A}_k$, $B$ satisfying Eq. (7) and $C = A$.*

One might ask whether such maximal rank solutions exist in the first place. The following proposition, whose proof is provided in the appendix, leverages Brouwer's fixed point theorem and the implicit function theorem to establish that, in general, they do exist.

**Proposition 4.7.** *Let $\mathbb{P}_0$ be some data distribution. For any positive $\epsilon$, there always exists a perturbed data distribution $(\tilde{x}, \tilde{y}) \sim \mathbb{P}_\epsilon$ that is $\epsilon$-close to $\mathbb{P}$ (e.g., in the 2-Wasserstein distance sense), so that there exist equilibria of the SG and EMA algorithms with $A$ having maximal rank $m$ for $\lambda$ small enough.*

Using classical results on the dynamics of differential equations will now allow us to prove our last result. Let us first define the stable equilibria of such a dynamical system (Arnol'd, 1992).

**Definition 4.8.** *An equilibrium for the dynamical system $\dot{z} = v(z)$, where $v$ is a smooth field over $\mathbb{R}^d$, is a point $e$ where $v(e) = 0$. An equilibrium $e$ is called (Lyapunov) stable when solutions of the differential equation with initial values close to $e$ converge uniformly to a nearby point. A stable equilibrium is said to be asymptotically stable when any solution started close to $e$ converges to $e$.*

**Theorem 4.9.** *(Arnol'd, 1992, Chap. 3) Consider a dynamical system $\dot{x} = v(x)$ whose dynamics can be approximated by the linear operator $J$: $v(x) = Jx + O(\|x\|^2)$. A sufficient condition for an equilibrium to be asymptotically stable is that all eigenvalues of $J$ have a negative real part.*

Armed with this classical result, we prove in the appendix the following proposition.

**Proposition 4.10.** *The equilibria of the dynamical system associated with the SG or the EMA procedure, if any, are, in general, asymptotically stable.*

Note that Proposition 4.10 does not imply that a nontrivial equilibrium exists or that the dynamics converge to such an equilibrium. It is, however, valid even in the case where $\alpha = 1$.

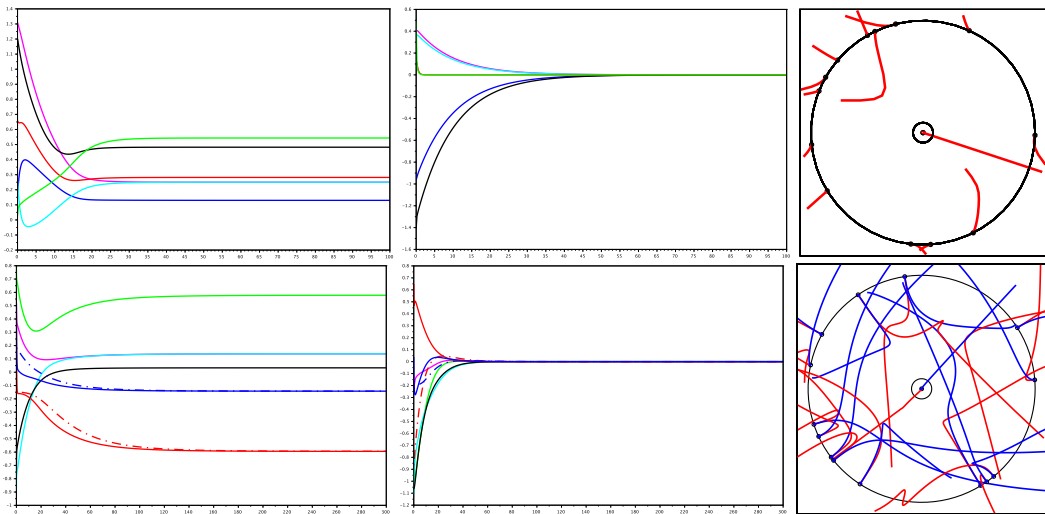

Figure 3: Integration paths for the SG (top) and EMA (bottom) procedures. In the top (resp. bottom) part of the figure, the left diagram shows a sample path for the 8 (resp. 6) coefficients of $a, B, c$ (resp. $a, B$), the central one shows a path where all parameters converge toward 0, and the right diagram shows 15 trajectories in the $a$ (resp. $a, c$) plane, starting from random locations. One of the trajectories has the origin as a limit point.

### 4.3 EXPERIMENTS WITH SYNTHETIC DATA

For these experiments, we consider the case of a scalar input space ($m = 1$) since $A$ is now a vector, which simplifies visualizations and numerical simulations, and denote by $\rho$ and $\tau$ the scalars $[xx^T]$ and $[yx^T]$. To emphasize that these quantities are vectors, let us use $a$ for $A$ and $c$ for $C$. In this simplified setting, equilibria are characterized by the following proposition.

**Proposition 4.11.** *In the case $m = 1$, a necessary and sufficient condition for the existence of nonzero equilibria of the SG and EMA algorithms is that $\Delta = \tau^2 - 4\rho\lambda \geq 0$. When this condition is satisfied, these equilibria are the pairs $(a, B)$ such that $a$ lies on either one of the hyperspheres $S_1$ and $S_2$ of $\mathbb{R}^n$ centered at the origin with radii $r_1 = (|\tau| - \sqrt{\Delta})/2\rho$ and $r_2 = (|\tau| + \sqrt{\Delta})/2\rho$, and $B$ verifies Eq. (7). The equilibria associated with $S_2$ are asymptotically stable, but those associated with $S_1$ are saddle points. The equilibria $(a, B, c)$ of the EMA algorithm and their stability can be characterized in a similar fashion, with the additional condition that $c = a$.*

Proposition 4.11 shows that the case $m = 1$ is both illustrative of the general case because the two spheres $S_1$ and $S_2$ are just the algebraic varieties identified in Proposition 4.5 and Corollary 4.6, and extremely nongeneric since the equilibria associated with $S_1$ are saddle points, which never happens when $m > 1$. See the proof of Proposition 4.11 in the appendix for an explanation of this phenomenon.

**Results and conclusions.** Figure 3 shows sample trajectories obtained by numerical simulations for $n = 2$, so $a$ and $c$ are points in the plane, and equilibria are located at the origin and on two circles centered at the origin with radii $r_1 < r_2$. We have (arbitrarily) taken $\rho = 3$, $\tau = 2$ and $\lambda = 0.1$ with $T = 300$ time steps. For the SG algorithm (Figure 3, top), we show on the left the evolution of $a$ (red and blue lines) and $B$ (in other colors), and in the middle a case where all coefficients converge to zero. Fifteen trajectories initialized from random positions and drawn in the $a$ plane are shown on the right. Figure 3 (bottom) illustrates the EMA algorithm. We use $\alpha_t = 0.9 + 0.1t/T$ in this experiment, so $\alpha = 1$ at $t = T$, following Grill et al. (2020). Although $c$ is not guaranteed to converge to $a$ in this case, it has done so in all our trials. This is illustrated on the left, where the $a$ parameters are shown as solid red and blue curves, and the $c$ parameters are shown as dashed red and blue curves (the other 4 curves correspond to $B$). The center part of the figure is an example where all parameters converge to zero. Sample trajectories starting from various random positions are shown on the right as red curves for $a$ and blue curves for $c$. Although we have not been able to prove the convergence of the EMA and SG algorithms so far, Figure 3 is typical of our observations: In 10,000 trials with parameters drawn at random,[4] the two algorithms always converge, 92.8%

---

[4]Values for $\rho$, $\tau$ and $\lambda$ are respectively drawn uniformly in $[0, 3]$, $[-1, 1]$ and $[0.01, 0.1]$. The coordinates of initial points are drawn from normal distributions with zero mean and unit variance.

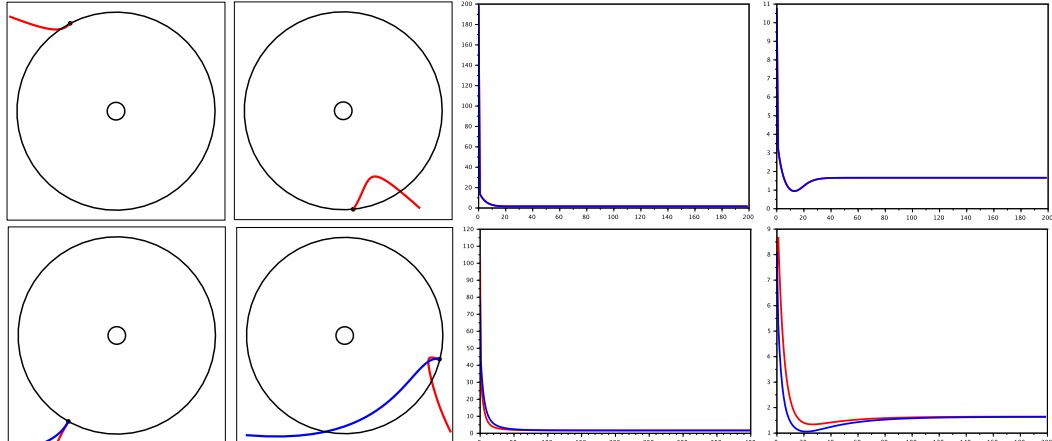

Figure 4: Top: Two trajectories for the SG algorithm plotted in the $a$ plane (left) and the evolution of $\bar{E}(a, B)$ values along these trajectories (right). The error appears to decrease steadily with time in the first case, but not in the second one. Bottom: A similar diagram for the EMA algorithm. In this case we show in red the curve $\bar{E}(a, B)$ and in blue the curve $\bar{F}(a, B, c)$.

(resp. 82.0%) to a limit point on the outer circle $S_2$ for EMA (resp. SG), and the rest of the time to the origin (as noted in Section 4.1, contrary to the generic nonlinear case, $A = 0$ and $B = 0$ are equilibria in the linear case). As expected, we have never observed convergence to the saddle points of $S_1$. As in the nonlinear case, the SG and EMA procedures do not appear to minimize $\bar{E}$ or $\bar{F}$. This is illustrated by Figure 4, where we plot the values of $\bar{E}(a, B)$ as a function of time for some trajectories of the SG and EMA algorithms. Although $\bar{E}(a, B)$ appears to steadily decrease in some cases, it definitely does not in other cases.

## 5    CONCLUSIONS

We have shown that the SG and EMA algorithms are not, in general, proper *optimization procedures*: In particular, they do not minimize *any* well defined objective function. On the other hand, they do not lead in general to collapse when they converge, and they enjoy interesting properties as *dynamical systems* since in the linear case any nontrivial limit point is asymptotically stable thus will not devolve into a trivial one by longer training. This point is important in practice since the SG and EMA training procedures empirically give good results, as shown for example in (Bardes et al., 2024; Grill et al., 2020; Chen and He, 2021), and appear, in the general nonlinear case, to prevent falling into the degenerate global minima they are designed to avoid. But then, what is it they really learn in the classical sense of the word? Much work remains to be done.

ETHICS STATEMENT

This work explores the theoretical foundations of two widely used methods in self-supervised learning, the stop-gradient and exponential moving average algorithms. Self-supervised representation learning has already had a significant impact in various applications, including natural language processing, computer vision, and robotics. We do not anticipate particular risks of this work, but a deeper understanding of self-supervised learning may allow the development of more performant and scalable algorithms, with potential increased impact on the aforementioned fields, which are not exempt of risks of misuse.

REPRODUCIBILITY STATEMENT

We have included in the main text all the necessary details to understand and reproduce our experimental results, both for the real-world data experiments, in Section 3.3, and for the synthetic data in Section 4.3. For the real-world data experiments, we use the publicly available code of V-JEPA (Bardes et al., 2024), which we slightly modify as explained in Section 3.3.

ACKNOWLEDGMENTS

This work was supported in part by the French government under management of Agence Nationale de la Recherche as part of the "France 2030" program, PR[AI]RIE-PSAI projet, reference ANR23-IACL-0008. JP was supported in part by the Louis Vuitton/ENS chair in artificial intelligence, the Institute of Information & Communications Technology Planning & Evaluation (IITP) grant funded by the Korean Government (MSIT) (No. RS-2024-00457882, National AI Research Lab Project), and a Global Distinguished Professorship at the Courant Institute of Mathematical Sciences and the Center for Data Science at New York University.

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

## 6 PROOFS

We will use, when convenient, the notation $f(\theta, x)$ for $f_\theta(x)$ and $g(\psi, z)$ for $g_\psi(z)$. For a given matrix $A$ we denote by $vec(A)$ the vector collecting all the components of the matrix $A$.

PROOF OF PROPOSITION 3.1

As noted before, $\bar{E}(\theta, \psi) = \bar{F}(\theta, \psi, \theta)$ for any values of $\theta$ and $\psi$. However, there is no a priori reason for a limit point of SG (if and when it converges) to be a critical point of $\bar{E}$. Let us write

$$F(\theta, \psi, \xi, x, y) = l[u(\theta, \psi, x), v(\xi, y)] + \Omega(\theta, \psi), \tag{9}$$

where $u(\theta, \psi, x) = g_\psi \circ f_\theta(x)$ and $v(\xi, y) = f_\xi(y)$, and denote by $J_\theta u(\theta, \psi, x)$ and $J_\psi u(\theta, \psi, x)$ the $n \times p$ and $n \times q$ Jacobians of $u(\theta, \psi, x) = g_\psi \circ f_\theta(x)$ with respect to $\theta$ and $\psi$. We also introduce $l(\theta, \psi, \xi, x, y) = l[u(\theta, \psi, x), v(\xi, y)]$. We have

$$\begin{cases} \nabla_\theta F = J_\theta u(\theta, \psi, x)^T \nabla_u l(\theta, \psi, \theta, x, y) + \nabla_\theta \Omega(\theta, \psi), \\ \nabla_\psi F = J_\psi u(\theta, \psi, x)^T \nabla_u l(\theta, \psi, \theta, x, y) + \nabla_\psi \Omega(\theta, \psi), \end{cases} \tag{10}$$

and the means of these two gradients (also gradients of $\bar{F}$) should vanish at any limit point of SG. But

$$\begin{cases} \nabla_\theta \bar{E}(\theta, \psi) = \nabla_\theta \bar{F}(\theta, \psi, \theta) + \mathbb{E}_{x,y}[J_\theta f(\theta, y)^T \nabla_v l[u(\theta, \psi, x), v(\theta, x, y)]], \\ \nabla_\psi \bar{E}(\theta, \psi) = \nabla_\psi \bar{F}(\theta, \psi, \theta), \end{cases} \tag{11}$$

where $J_\theta f(\theta, y) = J_\xi v(\theta, y)$ is the $n \times p$ Jacobian of $f$ with respect to $\theta$. There is a priori no reason why the second term of the gradient of $\bar{E}$ with respect to $\theta$, which depends on the data, should be zero at a critical point of $\bar{F}$ and thus at a limit point of SG (and thus of EMA) if such a point exists.

To establish this, we consider a counter example in the linear case where $f_\theta(x) = Ax$ and $g_\psi = Bz$ where $A$ is an $n \times m$ matrix and $B$ an $n \times n$ matrix with $n > m$. Here $\theta = vec(A)$ and $\xi = vec(B)$. Anticipating the result of Proposition 4.7, we know that we can always find an arbitrarily small perturbation to the data distribution, call it $\mathbb{P}_\epsilon$ so that there exists a matrix $A^\star$ of maximal rank and a matrix $B^\star$, that are critical points of the dynamics. Set $\theta^\star = vec(A^\star)$ and $\psi^\star = vec(B^\star)$. Hence, $f_{\theta^\star}(x)$ and $g_{\psi^\star}(z)$ cannot be degenerate solutions of the EMA or SG dynamics. Now, we wish to further establish that $\nabla_\theta \bar{E}_{\mathbb{P}_\epsilon}(\theta^\star, \psi^\star)$ is not 0 in a generic sense. Let us first express $\nabla_\theta \bar{E}_{\mathbb{P}_\epsilon}(\theta, \psi) := \mathbb{E}_{(\tilde{x}, \tilde{y}) \sim \mathbb{P}_\epsilon}[\nabla_\theta E(\theta, \psi, \tilde{x}, \tilde{y})]$ in terms of the matrices $A^\star$ and $B^\star$ and $\nabla_\theta \bar{F}_{\mathbb{P}_\epsilon}(\theta^\star, \psi^\star, \theta^\star) := \mathbb{E}_{(\tilde{x}, \tilde{y}) \sim \mathbb{P}_\epsilon}[\nabla_\theta F(\theta, \psi, \theta, \tilde{x}, \tilde{y})]$:

$$\nabla_\theta \bar{E}_{\mathbb{P}_\epsilon}(\theta^\star, \psi^\star) = \nabla_\theta \bar{F}_{\mathbb{P}_\epsilon}(\theta^\star, \psi^\star, \theta^\star) + A^\star[\tilde{y}\tilde{y}^\top] - B^\star A^\star[\tilde{x}\tilde{y}^\top]$$
$$= A^\star[\tilde{y}\tilde{y}^\top] - B^\star A^\star[\tilde{x}\tilde{y}^\top].$$

Note that $\nabla_\theta \bar{F}_{\mathbb{P}_\epsilon}(\theta, \psi, \theta)$ and $\nabla_\psi \bar{F}_{\mathbb{P}_\epsilon}(\theta, \psi, \theta)$ are independent of the matrix $[\tilde{y}\tilde{y}^\top]$ by virtue of the expression:

$$\nabla_\theta \bar{F}_{\mathbb{P}_\epsilon}(\theta, \psi, \theta) = B^\top(BA[\tilde{x}\tilde{x}^\top] - A[\tilde{y}\tilde{x}^\top]) + \lambda A$$
$$\nabla_\psi \bar{F}_{\mathbb{P}_\epsilon}(\theta, \psi, \theta) = BA[\tilde{x}\tilde{x}^\top]A^\top - A[\tilde{y}\tilde{x}^\top]A^\top + \lambda B.$$

Therefore, the critical points $(A^\star, B^\star)$ are independent of the matrix $[\tilde{y}\tilde{y}^\top]$. We treat two cases: either $\nabla_\theta \bar{E}_{\mathbb{P}_\epsilon}(\theta^\star, \psi^\star) \neq 0$, and there is nothing to prove, or $\nabla_\theta \bar{E}_{\mathbb{P}_\epsilon}(\theta^\star, \psi^\star) = 0$. In this case, we will construct a second perturbed data distribution $\mathbb{Q}_\epsilon$ from $\mathbb{P}_\epsilon$ as follows:

$$x' = \tilde{x} \qquad y' = \tilde{y} + \epsilon^{\frac{1}{2}}z, \tag{12}$$

where $(\tilde{x}, \tilde{y})$ are samples from $\mathbb{P}_\epsilon$ and $z$ is a standard centered gaussian independent of $\tilde{x}$ and $\tilde{y}$. It is easy to check that $[x'(x')^\top] = [\tilde{x}\tilde{x}^\top]$ and that $[y'(x')^\top] = [\tilde{y}\tilde{x}^\top]$ so that the critical points $A^\star$ and $B^\star$ remain unchanged. On the other hand $[y'(y')^\top] = [\tilde{y}\tilde{y}^\top] + \epsilon \mathrm{Id}$. It follows, under such perturbed distribution $\mathbb{Q}_\epsilon$ that:

$$\nabla_\theta \bar{E}_{\mathbb{Q}_\epsilon}(\theta^\star, \psi^\star) = \nabla_\theta \bar{F}_{\mathbb{Q}_\epsilon}(\theta^\star, \psi^\star, \theta^\star) + A^\star[y'(y')^\top] - B^\star A^\star[[x'(y')^\top]$$
$$= A^\star[\tilde{y}\tilde{y}^\top] - B^\star A^\star[\tilde{x}\tilde{y}^\top] + \epsilon A^\star$$
$$= \epsilon A^\star \neq 0.$$

Hence, we have established that one can always find a small perturbation $\mathbb{Q}_\epsilon$ to the data distribution so that the equilibrium is not degenerate and does not correspond to a critical point of the objective $\mathbb{E}_{(x',y') \sim \mathbb{Q}_\epsilon}[E(\theta, \psi, x', y')]$. $\qquad \square$

PROOF OF PROPOSITION 3.2

Define for simplicity, the following vector fields:

$$P(\theta, \psi, x, y) = \nabla_\theta F(\theta, \psi, \theta, x, y) \tag{13}$$
$$Q(\theta, \psi, x, y) = \nabla_\psi F(\theta, \psi, \theta, x, y). \tag{14}$$

Let $\mathbb{D}$ be some probability distribution over data $(x, y)$ and define $\bar{P}_{\mathbb{D}}(\theta, \psi) := \mathbb{E}_{(x,y)\sim\mathbb{D}}[P(\theta, \psi, x, y)]$ and $\bar{Q}_{\mathbb{D}}(\theta, \psi) := \mathbb{E}_{(x,y)\sim\mathbb{D}}[Q(\theta, \psi, x, y)]$. According to Schwarz's integrability theorem, a *necessary* condition for $\bar{P}_{\mathbb{D}}$ and $\bar{Q}_{\mathbb{D}}$ to be the gradient field of a smooth scalar function is that their second-order cross derivatives be the transposes of each other. We have

$$\begin{cases} \dfrac{\partial P}{\partial \psi}(\theta, \psi, x, y) = H_{\theta,\psi} u(\theta, \psi, x)[u(\theta, \psi, x) - v(\theta, y)] + J_\theta u(\theta, \psi, x)^T J_\psi u(\theta, \psi, x), \\[2mm] \dfrac{\partial Q}{\partial \theta}(\theta, \psi, x, y) = H_{\psi,\theta} u(\theta, \psi, x)[u(\theta, \psi, x) - v(\theta, y)] + J_\psi u(\theta, \psi, x)^T [J_\theta u(\theta, \psi, x) - J_\theta v(\theta, y)], \end{cases} \tag{15}$$

where $J_\theta v(\theta, y)$ denotes the $n \times p$ Jacobian of $v$ with respect to $\theta$, which is equal to the Jacobian $J_\theta f(\theta, y)$, and $H_{\theta,\psi} u$ (resp. $H_{\psi,\theta} u$) is the $p \times q \times n$ (resp. $q \times p \times n$) tensor associated with the second partial derivative of $u$ with respect to $\theta$ and $\psi$ (resp. $\psi$ and $\theta$). The products of these tensors by the vector $u - v$ yield $p \times q$ and $q \times p$ matrices that are transposes of each other, and we obtain

$$\bar{\Delta}_{\mathbb{D}}(\theta, \psi) \stackrel{\text{def}}{=} \frac{\partial \bar{P}_{\mathbb{D}}}{\partial \psi}(\theta, \psi) - \frac{\partial \bar{Q}_{\mathbb{D}}}{\partial \theta}(\theta, \psi)^T = \mathbb{E}_{x,y}[\underbrace{J_\theta f(\theta, y)^T J_\psi g(\psi, f(\theta, x))}_{\Delta(\theta, \xi, x, y)}]. \tag{16}$$

We wish to establish that $\bar{\Delta}_{\mathbb{D}} \neq 0$ in a generic sense over all possible data distributions. This means that even if for some data distribution $\mathbb{D}_0$ we have $\bar{\Delta}_{\mathbb{D}_0} = 0$, we can always construct a perturbed data distribution $\mathbb{D}_\epsilon$ that gets arbitrarily close to $\mathbb{D}_0$ as $\epsilon$ approaches 0 and for which $\bar{\Delta}_{\mathbb{D}_\epsilon} \neq 0$. To this end, let's consider a case for which $\bar{\Delta}_{\mathbb{D}_0} = 0$ for some $\mathbb{D}_0$, otherwise there is nothing to prove.

By Proposition 6.1 stated below, we know that the integrand $\Delta$ appearing in the expression of $\bar{\Delta}_{\mathbb{D}_0}$ is a function that is not identically 0. Hence, there exists $\theta, \xi, x_0$ and $y_0$ so that $\|\Delta(\theta, \xi, x_0, y_0)\|_F \geq 2\eta$ for some positive $\eta$. Since both function $f$ and $g$ are assumed to be continuously differentiable, we know that $\Delta$ must be a continuous map. Hence, there exists a positive radius $r > 0$ small enough so that $\|\Delta(\theta, \xi, x, y) - \Delta(\theta, \xi, x_0, y_0)\|_F \leq \eta$ for any $(x, y)$ in a ball $\mathcal{B}$ centered around $(x_0, y_0)$ of radius $r$. Consider uniform distribution $q$ over such ball. It is then easy to see that $\|\mathbb{E}_{(x,y)\sim q}[\Delta(\theta, \xi, x, y)] - \Delta(\theta, \xi, x_0, y_0)\|_F \leq \eta$, so that:

$$\eta \leq \|\Delta(\theta, \xi, x_0, y_0)\|_F - \|\mathbb{E}_{(x,y)\sim q}[\Delta(\theta, \xi, x, y)] - \Delta(\theta, \xi, x_0, y_0)\|_F. \tag{17}$$

Now, define the following perturbed distribution by convex mixture of $\mathbb{D}_0$ and $q$:

$$\mathbb{D}_\epsilon = (1 - \epsilon)\mathbb{D}_0 + \epsilon q. \tag{18}$$

We will show that $\|\bar{\Delta}_{\mathbb{D}_\epsilon}\|_F \geq \epsilon\eta$. To see this, we first notice that:

$$\bar{\Delta}_{\mathbb{D}_\epsilon} = (1 - \epsilon)\underbrace{\bar{\Delta}_{\mathbb{D}_0}}_{=0} + \epsilon\mathbb{E}_{(x,y)\sim q}[\Delta(\theta, \xi, x, y)]. \tag{19}$$

Hence, by direct application of the triangle inequality, we have that:

$$\epsilon\eta \leq \epsilon\left(\|\Delta(\theta, \xi, x_0, y_0)\|_F - \|\mathbb{E}_{(x,y)\sim q}[\Delta(\theta, \xi, x, y)] - \Delta(\theta, \xi, x_0, y_0)\|_F\right) \leq \|\bar{\Delta}_{\mathbb{D}_\epsilon}\|_F. \tag{20}$$

We have, therefore shown an arbitrarily small perturbation on the data distribution suffices to ensure that $\bar{\Delta}_{\mathbb{D}_\epsilon} \neq 0$, which in turn implies that that a the corresponding vector fields $\bar{P}_{\mathbb{D}_\epsilon}$ and $\bar{Q}_{\mathbb{D}_\epsilon}$ are not the gradient of any scalar function. $\square$

The interested reader may wonder how the integrability condition of Proposition 3.2 translates in the linear case. As shown below, a necessary condition for $\bar{P}$ and $\bar{Q}$ to be the gradient field of a smooth function for linear encoders and predictors is that $A[xy^T] = 0$. This condition depends on the data and is not in general satisfied by $A$, as expected.

The proof goes as follows: Given any $s \times t$ matrix $U$ with rows $u_i^T$ ($i = 1, \ldots, s$) and vector $v$ in $\mathbb{R}^t$, let $(Uv)_i = u_i \cdot v$ denote the $i$th entry of $Uv$. We have $\partial(Uv)_i/\partial u_i = v$ and $\partial(Uv)_i/\partial u_j = 0$

for any $j \neq i$, and it follows that the two Jacobians $J_\theta(Ay)$ and $J_\psi(BAx)$ are respectively the $n \times p$ and $n \times q$ matrices

$$
\begin{bmatrix} y^T & \dots & 0^T \\ \vdots & \ddots & \vdots \\ 0^T & \dots & y^T \end{bmatrix} \text{ and } \begin{bmatrix} x^T A^T & \dots & 0^T \\ \vdots & \ddots & \vdots \\ 0^T & \dots & x^T A^T \end{bmatrix}. \tag{21}
$$

Substituting these values in Eq. (16) shows that $\Delta(\theta, \psi)$ is a $p \times q$ block diagonal matrix whose $n$ ($m \times n$) diagonal blocks are all equal to $[yx^T]A^T$. Transposing $\Delta(\theta, \psi) = 0$ concludes the proof.

We end this section with the main proposition used in the proof above.

**Proposition 6.1.** *Assume that the parametric encoder and predictor are not identically* 0, *i.e. there exists* $\theta_0, \psi_0, x_0$ *and* $z_0$ *so that* $f(\theta_0, x_0) \neq 0$ *and* $g(\psi_0, z_0) \neq 0$. *Furthermore, assume both* $f$ *and* $g$ *have a final linear layer, In other words, they can be expressed in the following form:*

$$
f(\theta, x) := A\phi(U, x), \qquad g(\psi, z) := Bh(V, z) \tag{22}
$$

*where* $A$ *and* $B$ *are* $n \times k$ *and* $n \times d$ *matrices for some positive integers* $k$ *and* $d$, *while* $U$ *and* $V$ *are parameters so that* $\theta = vec(A, U)$ *and* $\psi = vec(B, V)$. *Here,* $\phi$ *and* $h$ *are differentiable functions with values in* $\mathbb{R}^k$ *and* $\mathbb{R}^d$. *Furthermore, consider the quantity*

$$
\Delta(\theta, \psi, x, y) := J_\theta f(\theta, y)^\top J_\psi g(\psi, f(\theta, x)), \tag{23}
$$

*where* $J_\theta f$ *and* $J_\psi g$ *denote the Jacobians of* $f$ *and* $g$ *w.r.t.* $\theta$ *and* $\psi$. *Then* $\Delta(\theta, \psi, x, y)$ *cannot be identically* 0.

*Proof.* It suffices to show that the components $\tilde{\Delta}$ of $\Delta$ corresponding to the partial derivatives w.r.t. the linear parameters $A$ and $B$ are not identically 0. Hence, by contradiction, we will assume that $\tilde{\Delta}$ vanishes everywhere. Hence, $\tilde{\Delta}$ must vanish, when applied to any arbitrary perturbation matrices $\delta A$ and $\delta B$ of $A$ and $B$, so that the following holds:

$$
\tilde{\Delta}(\theta, \psi, x, y)(\delta A, \delta B) = (\delta A\phi(U, y))^\top \delta Bh(V, A\phi(U, x)) = 0. \tag{24}
$$

The above identity is obtained by standard calculus. Furthermore, since $f$ and $g$ are non vanishing, there exists parameter values $U_0$ and $V_0$ so that $x \mapsto \phi(U_0, x)$ and $z \mapsto h(V_0, z)$ are not identically 0. Let $y_0$ be a vector in $\mathbb{R}^m$ so that $\phi(U_0, y_0) \neq 0$. Moreover, fix any arbitrary vector $z \in \mathbb{R}^n$. There must exist a perturbation matrix $\delta A$ so that $z = \delta A\phi(U_0, y_0)$ (simply take $\delta A = \frac{1}{\|\phi(U_0, y_0)\|^2}z\phi(U_0, y_0)^\top$). Therefore, by Equation 24 it follows:

$$
z^\top(\delta Bh(V, A\phi(U_0, x))) = 0, \tag{25}
$$

for any $z$ in $\mathbb{R}^n$ and any matrix $\delta B$. This directly implies that $h(V, A\phi(U_0, x)) = 0$ for any $A$, $x$ and $V$. Furthermore, by choosing $V = V_0$, $x = y_0$ and $A = \frac{1}{\|\phi(U_0, y_0)\|^2}z\phi(U_0, y_0)^\top$ for any arbitrary $z \in \mathbb{R}^n$, we get that: $h(V_0, z) = 0$. This contradicts the fact that $z \mapsto h(V_0, z)$ is not identically 0. Hence, we have shown that $\tilde{\Delta}$ is not identically 0 which, a fortiori, implies that $\Delta$ is itself not identically null. $\square$

PROOF OF LEMMA 4.1

We can rewrite $F$ in this case as

$$
\begin{aligned}
F(A, B, C, x, y) &= \tfrac{1}{2}\|BAx - Cy\|^2 + \tfrac{\lambda}{2}(\|A\|_F^2 + \|B\|_F^2), \\
&= \tfrac{1}{2}x^T A^T B^T BAx - y^T C^T BAx + \tfrac{1}{2}y^T C^T Cy + \tfrac{\lambda}{2}(\|A\|_F^2 + \|B\|_F^2).
\end{aligned} \tag{26}
$$

We know from Eqs. [70] and [82] in *the matrix cookbook* (Petersen and Pedersen, 2012) that

$$
\begin{cases} \frac{\partial}{\partial U}(a^T Ub) = ab^T, \\ \frac{\partial}{\partial U}(b^T U^T VUc) = V^T Ubc^T + VUcb^T. \end{cases} \tag{27}
$$

It follows that:

$$
\begin{cases} \frac{\partial F}{\partial A}(A, B, C, x, y) = B^T(BAx - Cy)x^T + \lambda A, \\ \frac{\partial F}{\partial B}(A, B, C, x, y) = (BAx - Cy)x^T A^T + \lambda B. \end{cases} \tag{28}
$$

$\square$

PROOF OF LEMMA 4.2

Multiplying the first equation in (4) by $A^T$ on the right and subtracting the second one multiplied by $B^T$ on the left immediately yields $B^T B - AA^T = 0$ at a limit point since both derivatives $\dot{P}(A, B, C)$ and $\dot{Q}(A, B, C)$ are zero at such a point. $\qquad\square$

PROOF OF LEMMA 4.3

Let us first write the derivative of $E$ with respect to $A$. We have

$$\frac{\partial E}{\partial A} = \frac{\partial F}{\partial A} + Ayy^T - BAxy^T = B^T(BAx - Ay)x^T - (BAx - Ay)y^T + \lambda A, \qquad (29)$$

and thus $\dot{A} = -(B^T R(A, B) - S(A, B) + \lambda A)$, where $S(A, B) = BA[xy^T] - A[yy^T]$. Substituting in the temporal derivative of $1/2\|A\|_F^2$, we obtain

$$
\begin{aligned}
\frac{d}{dt}[\frac{1}{2}\|A\|_F^2] &= \frac{d}{dt}[\frac{1}{2}\text{tr}(A^T A)] = -\text{tr}(A^T \dot{A}) = \text{tr}(A^T B^T R(A, B) + A^T S(A, B) + \lambda A^T A) \\
&= -\text{tr}(\mathbb{E}_{x,y}[x^T A^T B^T BAx + y^T A^T Ay - x^T A^T B^T Ay - y^T A^T BAx] + \lambda A^T A) \\
&= -(\mathbb{E}_{x,y}[\|BAx - Ay\|^2] + \lambda\|A\|_F^2) \\
&\leq -\lambda\|A\|_F^2
\end{aligned}
$$

$$(30)$$

which implies the exponential convergence of $\|A\|_F^2$ toward zero. $\qquad\square$

PROOF OF LEMMA 4.4

Multiplying on both sides the first equality in Eq. (5) on the right by $A^T$ and the second one on the left by $B^T$ and taking the difference yields

$$B^T(\dot{B} + \lambda B) = (\dot{A} + \lambda A)A^T \qquad (31)$$

Adding this equation to its transpose and multiplying both sides by $e^{2\lambda t}$ now yields

$$e^{2\lambda t}(\dot{B}^T B + B^T \dot{B} + 2\lambda B^T B) = e^{2\lambda t}(A\dot{A}^T + \dot{A}A^T + 2\lambda AA^T), \qquad (32)$$

from which we conclude that

$$\frac{d}{dt}[e^{2\lambda t}B^T B] = \frac{d}{dt}[e^{2\lambda t}AA^T], \qquad (33)$$

and obtain $AA^T = B^T B + e^{-2\lambda t}K$, where $K$ is a constant independent of the data. This implies in particular that $B^T B - AA^T \to 0$ as $t \to +\infty$. $\qquad\square$

PROOF OF PROPOSITION 4.5

We only give there the proof for the SG algorithm since the proof for the EMA algorithm follows the exact same reasoning with the extra parameter $C$ known to be equal to $A$ at an equilibrium.

The equilibria $(A, B)$ of the SG algorithm are characterized by the two equations

$$
\begin{cases}
0 = B^T(BA[xx^T] - A[yx^T]) + \lambda A, \\
0 = (BA[xx^T] - A[yx^T])A^T + \lambda B = BW - A[yx^T]A^T = 0.
\end{cases}
\qquad (34)
$$

The fact that $B^T B = AA^T$ follows immediately from multiplying both sides of the first condition by $A^T$ on the right and both sides of the second one by $B^T$ on the left, then taking the difference. Equation (7) follows immediately from the second condition, the inverse being well defined when $\lambda > 0$ since $W$ is symmetric positive definite in this case. Substituting this value in the first equality of Eq. (34) now yields

$$W^{-1}A[xy^T]A^T \left(A[yx^T]A^T W^{-1}A[xx^T] - A[yx^T]\right) + \lambda A = 0, \qquad (35)$$

and equilibria of the SG algorithm are exactly the pairs $(A, B)$ where $A$ satisfies this condition and $B$ is given by Eq. (7). Assuming that $A$ has full rank $m$ and multiplying both sides of this equation on the left by $W$ and on the right by $A^T$ yields the equivalent condition

$$A[xy^T]A^T \left(A[yx^T]A^T W^{-1}(W - \lambda\text{Id}) - A[yx^T]A^T\right) + \lambda WAA^T = 0. \qquad (36)$$

Multiplying both sides of this equation on the right by $W$, we obtain the condition

$$A(U^T U - V^T V)A^T = 0, \text{ where } \begin{cases} U = A^T A[xx^T] + \lambda \text{Id}, \\ V = A[yx^T], \end{cases} \tag{37}$$

which is equivalent to $U^T U = V^T V$ since we have assumed that $A$ has full rank, and thus to Eq. (6) as well. $\qquad \square$

Note that Tian et al. (2021) give in Appendix D of their paper an alternative characterization of the equilibria in the case where $\lambda = 0$, which essentially corresponds to the condition of Eq. (7) in this case, *without* the characterization of $S = A^T A$ by Eq. (6).

PROOF OF COROLLARY 4.6

The proof is textbook material and included for completeness. When $U$ is column orthogonal and $A = U\sqrt{S_k}$, we obviously have $A^T A = S_k$. Conversely, when $A^T A = S_k$, let us take $U = A\sqrt{S_k}^{-1}$ (the inverse is guaranteed to exist since $S_k$ is positive definite). We have $U^T U = \sqrt{S_k}^{-1} S_k \sqrt{S_k}^{-1} = \text{Id}$. $\qquad \square$

PROOF OF PROPOSITION 4.7

Fix $\epsilon > 0$. There exist $0 < \delta \le \epsilon$ positive so that $T := [yx^\top] + \delta I$ and $R := [xx^\top] + \delta I$ are both invertible. To see this, it suffices to notice that $\delta \mapsto det([yx^\top] + \delta I)$ is a non-zero polynomial, thus vanishes for a finite number of values $\delta$. Therefore, we can always find $\delta < \epsilon$ for which $det([yx^\top] + \delta I) \ne 0$, so that $T$ is invertible. Furthermore, $R$ is necessarily invertible since it is the sum of the PSD matrix $[xx^\top]$ and the PD matrix $\delta I$. The matrices $R$ and $T$ correspond to the covariances of the following perturbed variables:

$$\tilde{x} = x + \delta^{\frac{1}{2}} z, \qquad \tilde{y} = y + \delta^{\frac{1}{2}} z, \tag{38}$$

where $z$ is a standard isotropic gaussian. Thus we have $R = [\tilde{x}\tilde{x}^\top]$ and $T = [\tilde{y}\tilde{x}^\top]$. We will show that there exists a positive $\lambda_0$ small enough so that the following equation always admits a PD solution for any $0 \le \lambda \le \lambda_0$:

$$([\tilde{x}\tilde{x}^\top]S + \lambda Id)(S[\tilde{x}\tilde{x}^\top] + \lambda Id) = [\tilde{y}\tilde{x}^\top]S[\tilde{y}\tilde{x}^\top]. \tag{39}$$

We will first establish existence of the solution for $\lambda = 0$, then show that the property still holds for $\lambda$ small enough.

**Case $\lambda = 0$.** In this case, the equation simplifies to:

$$RS^2 R = T^\top ST, \tag{40}$$

where we used the notation $R$ and $T$ for simplicity. Since, $R$ is invertible, we multiply both sides by $R^{-1}$ (left and right), which yields:

$$S^2 = (TR^{-1})^\top S(TR^{-1}). \tag{41}$$

Since the matrix $TR^{-1}$ is invertible, we can directly apply the technical Lemma 6.2, stated below, which guarantees the existence of a PD solution $S^\star$ to the above equation.

**Case $\lambda > 0$.** We will apply the implicit function theorem to show the existence of solutions for $\lambda$ small enough. Consider the matrix valued map $\mathcal{G}(\lambda, S)$ defined as:

$$\mathcal{G}(\lambda, S) = S^2 + \lambda(SR^{-1} + R^{-1}S) + \lambda^2 Id - (TR^{-1})^\top S(TR^{-1}). \tag{42}$$

We have already established that the equation $\mathcal{G}(0, S) = 0$ admits a solution $S^\star$. It suffices to prove that the partial differential $d_S\mathcal{G}(0, S^\star)$ at $(0, S^\star)$ is invertible. Since $d_S\mathcal{G}(0, S^\star)$ is a linear map from the set of $m \times m$ matrices to itself, it suffices to establish its injectivity. Direct calculations show that $H \mapsto d_S\mathcal{G}(0, S^\star)(H)$ is given by:

$$d_S\mathcal{G}(0, S^\star)(H) = S^\star H + HS^\star - (TR^{-1})^\top H(TR^{-1}). \tag{43}$$

Using again the technical Lemma 6.2, we know that the only solution to the equation $d_S\mathcal{G}(0, S^\star)(H) = 0$ is $H = 0$. Hence, $d_S\mathcal{G}(0, S^\star)$ is injective. Therefore, by the implicit function theorem, there exists a positive $\lambda_0$ so that for any $0 \le \lambda \le \lambda_0$, the equation $\mathcal{G}(\lambda, S) = 0$ admits a solution $S^\star$.

**Lemma 6.2.** *Let $C$ be an invertible $m \times m$ matrix. There exists a symmetric PD solution $X^\star$ to the following equation:*

$$X^2 = C^\top X C.$$

*Moreover, consider the following linear system $X^\star H + H X^\star - C^\top H C = 0$ with unknown $H$. The only solution to such system is $H = 0$.*

*Proof.* **Existence.** We will apply Brouwer's fixed point theorem to a suitable operator. Denote by $\mu$ and $\rho$ its smallest and largest eigenvalues of $C^\top C$ which are positive. Define the following continuous map $\mathcal{G}$ over the set of $\mathbb{S}_m^+$ of symmetric PSD matrices of size $m \times m$:

$$\mathcal{G}(X) = (C^\top X C)^{\frac{1}{2}},$$

where the square root denotes the unique PSD square root of a PSD matrix. Note that $\mathcal{G}(X)$ is well defined for any $X \in \mathbb{S}_m^+$. Consider the following set of matrices:

$$\mathbb{M} = \{X \in \mathbb{S}_m^+ : \quad \mu I \leq X \leq \rho I\}.$$

Then $\mathbb{M}$ is a convex compact subset of vector space of $n \times n$ matrices. We will show that $\mathcal{G}(\mathbb{M}) \subset \mathbb{M}$, which will allow us to apply Brouwer fixed point theorem. Indeed, for any $X$ in $\mathbb{M}$, simple matrix inequalities yield:

$$\mu C^\top C \leq C^\top X C \leq \rho C^\top C.$$

Recalling that the symmetric square root preserves the matrix order, we directly get:

$$\mu^{\frac{1}{2}} (C^\top C)^{\frac{1}{2}} \leq \mathcal{G}(X) \leq \rho^{\frac{1}{2}} (C^\top C)^{\frac{1}{2}}.$$

However, by definition of $\rho$ and $\mu$, we have that $\mu I \leq C^\top C \leq \rho I$. Consequently, it follows that $\mu I \leq \mathcal{G}(X) \leq \rho I$. Hence, we have established that $\mathbb{M}$ is a stable set of the map $\mathcal{G}$. Since the map $\mathcal{G}(X)$ is continuous and $\mathbb{M}$ is a convex compact subset of the space of square matrices, it follows by Brouwer's fixed point theorem that there exists $X^\star$ satisfying the equation $\mathcal{G}(X^\star) = X^\star$. After taking the square of such equation, we get that $X^\star$ is a solution to $X^2 = C^\top X C$.

**Uniqueness of the solution to the linear system.** Let $H$ be an $m \times m$ matrix solution to the linear system:

$$X^\star H + H X^\star - C^\top H C = 0.$$

We wish to show that $H = 0$. We can multiply such equation (left and right) by $(X^\star)^{-\frac{1}{2}}$ to get:

$$(X^\star)^{\frac{1}{2}} H (X^\star)^{-\frac{1}{2}} + (X^\star)^{-\frac{1}{2}} H (X^\star)^{\frac{1}{2}} - (X^\star)^{-\frac{1}{2}} C^\top H C (X^\star)^{-\frac{1}{2}} = 0$$

Define $E = (X^\star)^{-\frac{1}{2}} C^\top (X^\star)^{\frac{1}{2}}$. By direct calculation and using the definition of $X^\star$ (i.e. the solution to the equation $X^2 = C^\top X C$), we have that $E$ satisfies $E E^\top = X^\star$. Now, consider the change of variables $\tilde{H} = (X^\star)^{-\frac{1}{2}} H (X^\star)^{-\frac{1}{2}}$. We can thus express the above equation in terms of $\tilde{H}$ and $E$ and $X^\star$ as follows:

$$X^\star \tilde{H} + \tilde{H} X^\star - E \tilde{H} E^\top = 0.$$

Since $E$ is invertible, we can further multiply the equation by $E^{-1}$ on the left and by $E^{-\top}$ on the right and use the identity $E E^\top = X^\star$ to get:

$$E^\top \tilde{H} E^{-\top} + E^{-1} \tilde{H} E = \tilde{H}.$$

The above equation directly implies that $\tilde{H}$ must be symmetric. Furthermore, by direct calculation, we obtain the following expression for $\|\tilde{H}\|_F^2$:

$$\begin{aligned}
\|\tilde{H}\|_F^2 &= Tr\left((E^\top \tilde{H} E^{-\top} + E^{-1} \tilde{H} E)^\top (E^\top \tilde{H} E^{-\top} + E^{-1} \tilde{H} E)\right) \\
&= \mathrm{tr}(E^T \tilde{H}^2 E^{-\top}) + 2\mathrm{tr}(E^\top \tilde{H} E^{-\top} E^{-1} \tilde{H} E) + \mathrm{tr}(E^{-1} \tilde{H}^2 E) \\
&= 2\|\tilde{H}^2\|_F^2 + 2\|E^{-1} \tilde{H} E\|_F^2.
\end{aligned}$$

The above identity can only be true if $\|\tilde{H}\|_F^2 = 0$ which directly implies that $H = 0$ since $X^\star$ is invertible.

$\square$

PROOF OF PROPOSITION 4.10

We only present here the proof for the EMA procedure. The SG case is similar and slightly simpler, and thus omitted for conciseness. Let $(A, B, C)$ be an equilibrium point of our dynamical system and $(A + D, B + E, C + F)$ a point in its neighborhood, where, like $A$, $B$ and $C$, the matrices $D$, $E$ and $F$ are respectively of size $n \times m$, $n \times n$ and $n \times m$. To first order, we have

$$\begin{cases} \dot{A}(A + D, B + E, C + F) \approx -B^T((EA + BD)[xx^T] - F[yx^T]) - E^T R(A, B, C) - \lambda D, \\ \dot{B}(A + D, B + E, C + F) \approx -((EA + BD)[xx^T] - F[yx^T])A^T - R(A, B, C)D^T - \lambda E, \\ \dot{C}(A + D, B + E, C + F) = (1 - \alpha)(D - F). \end{cases}$$
(44)

The eigenvectors of the corresponding linear operator in $D$, $E$ and $F$ and the corresponding eigenvalues $\mu$ are thus characterized by

$$\begin{cases} 0 = B^T((EA + BD)[xx^T] - F[yx^T]) + E^T R(A, B, C) + (\lambda + \mu)D, \\ 0 = ((EA + BD)[xx^T] - F[yx]^T)A^T + R(A, B, C)D^T + (\lambda + \mu)E, \\ (1 - \alpha)D = (1 - \alpha + \mu)F. \end{cases}$$
(45)

Let us first consider the trivial equilibrium where $A = C = 0$ and $B = 0$. Substituting these values in Eq. (45) shows that for any triplet $(D, E, F)$ satisfying this equation we must have either $\mu = -\lambda < 0$ if $D$ or $E$ is nonzero, or $\mu = \alpha - 1$ if $D$ and $E$ are both zero. But, as noted before, $\alpha$ is normally taken smaller than or equal to 1 and we assume here that $\alpha \neq 1$ (the moving average would not make much sense otherwise since $\xi_t$ would be constant in that case), so $\mu < 0$ in that case as well. It follows that trivial equilibria are asymptotically stable

Let us now consider the case of nontrivial equilibria where, in particular, $A \neq 0$. Note that for any eigenvector triplet $(D, E, F)$ and associated eigenvalue $\mu$ satisfying Eq. (45), either $1 - \alpha + \mu$ is zero, in which case $F$ can take any value and, as just observed, $\mu < 0$, or it is not, in which case $F = \beta D$ with $\beta = (1 - \alpha)/(1 - \alpha + \mu)$ so we can focus on the first two equations.

Multiplying the first one on the right by $A^T$ and the second one on the left by $B^T$, subtracting the two and using Eq. (34) yields

$$(\lambda + \mu)M = -\lambda M^T \text{ where } M = (DA^T - B^T E).$$
(46)

Now, any matrix $U$ such that $U^T = aU$ for some scalar $a$ also verifies $U = aU^T$ by taking the transpose on both sides, and thus $U = aU^T = a(aU) = a^2 U$, which means that, either $U = 0$ or $U \neq 0$ and $a^2 = 1$, with either $a = 1$ and $U$ symmetric or $a = -1$ and $U$ skew-symmetric. In our case, when $M \neq 0$, it is either symmetric with $\mu = -2\lambda$, or skew-symmetric with $\mu = 0$. In the first case, any equilibrium is asymptotically stable according to Theorem 4.9 while, in the second one, all eigenvalues vanish and nothing can be said to first order, which should not happen generically.

Let us now prove that $\mu < 0$ in the slightly more complicated case $M = 0$. Using $B^T E = DA^T$, multiplying again the first equation in (45) on the right by $A^T$ (or the second one by $B^T$ on the left) and using Eq. (34) now yields

$$\begin{aligned} 0 &= B^T((EA + BD)[xx^T] - F[yx^T])A^T + E^T R(A, B, C)A^T + (\lambda + \mu)DA^T \\ &= (DA^T A + AA^T D)[xx^T] - B^T F[yx^T])A^T - \lambda AD^T + (\lambda + \mu)DA^T, \\ &= N + P + \mu DA^T, \end{aligned}$$
(47)

where

$$\begin{cases} N = AA^T D[xx^T]A^T - \beta B^T D[yx^T]A^T - \lambda AD^T, \\ P = DA^T A[xx^T]A^T + \lambda DA^T = DA^T(A[xx^T]A^T + \lambda \text{Id}). \end{cases}$$
(48)

Here, $N$ is an $n \times n$ matrix of rank at most $m$ with $n > m$. It is therefore singular with a kernel of dimension $n - m$. So is its transpose. Let us pick $u$ in $\text{Ker}(N^T)$ such that $AD^T u \neq 0$. Generically this is always possible since there is no reason for $\text{Ker}(N^T)$ and $\text{Ker}(AD^T)$ to coincide.

Multiplying the second equation in (48) on the left by $u^T$ and on the right by $v = AD^T u \neq 0$ now yields

$$v^T(A[xx^T]A^T + \lambda \text{Id})v + \mu\|v\|^2 = 0,$$
(49)

and since $A[xx^T]A^T + \lambda \text{Id}$ is positive definite, we conclude that $\mu < 0$. $\quad\square$

PROOF OF PROPOSITION 4.11

When $m = 1$, $S = \|a\|^2$ and Eq. (6) can be rewritten as

$$(\rho\|a\|^2 + \lambda)^2 = \tau^2\|a\|^2, \tag{50}$$

or equivalently

$$\rho\|a\|^2 - \varepsilon\tau\|a\| + \lambda = 0 \text{ where } \varepsilon = \mp 1. \tag{51}$$

A necessary and sufficient condition for real solutions of this quadratic equation in $\|a\|$ to exist is that its discriminant $\Delta$ be nonnegative and they are nonnegative when $\varepsilon\tau$ is itself nonnegative, i.e., $\varepsilon = \text{sign}(\tau)$, so

$$\rho\|a\|^2 - |\tau|\,\|a\| + \lambda = 0. \tag{52}$$

These solutions indeed correspond to the two hyperspheres $S_1$ and $S_2$ defined above, which concludes the proof of the first part of the proposition. These correspond exactly to the varieties $\mathcal{A}_1$ and $\mathcal{A}_2$ associated with the two positive roots $r_1^2$ and $r_2^2$ of the quadratic equation Eq. (6) in $\|a\|^2$ of course. Now, let $a$ be an element of $S_i$ ($i = 1, 2$). According to Eqs. (7) and (52), we have

$$B = \tau a a^T W^{-1} = \frac{\tau}{\rho\|a\|^2 + \lambda} a a^T = \frac{1}{r_i}\text{sign}(\tau)aa^T. \tag{53}$$

This concludes the first part of the proof of the proposition.

Let us now turn to its second part, assume $\Delta \geq 0$ and consider an equilibrium with $a$ in $S_i$. As shown in the proof of Prop. 4.10, generically, all eigenvalues are negative unless $M = da^T - B^T E = 0$. Substituting the value of $B$ in this equation shows that $da^T = (\text{sign}(\tau)/r_i)aa^T E$, and since eigenvectors are only defined up to scale we can pick $d = a$ and $E^T a = \text{sign}(\tau)r_i a$ (note that there exists an infinity of $n \times n$ matrices $E$ verifying this equality, including $E = B$). Substituting in Eq. (45) and using Eqs. (51) and (52) now yields

$$0 = (3\rho r_i^2 - 2\text{sign}(\tau)r_i + \lambda + \mu)a = (\mu - \lambda + \rho r_i^2)a = (\mu - 2\lambda + |\tau|\,r_i)a, \tag{54}$$

and thus, when $a \neq 0$, $\mu = 2\lambda - |\tau|r_i$ (note that with this value for $\mu$, $d = a$ and any matrix $E$ such that $E^T a = \text{sign}(\tau)r_i a$ satisfy Eq. (45) and are thus indeed the $(d, E)$ part of the corresponding eigenvector). Now, a $r_i = (|\tau| + \eta_i\sqrt{\Delta})/2\rho$ where $\eta_1 = -1$ and $\eta_2 = 1$, so we have

$$\mu = 2\lambda - \frac{1}{2\rho}|\tau|(|\tau| + \eta_i|\tau|\sqrt{\Delta}) = \frac{-1}{2\rho}(\Delta + \eta_i|\tau|\sqrt{\Delta}) = \frac{-\Delta}{2\rho}(\sqrt{\Delta} + \eta_i|\tau|). \tag{55}$$

In particular, a necessary and sufficient condition for $\mu$ to be positive is that $\eta_1 = -1$, corresponding to the hypersphere of radius $r_1$, and that

$$\tau^2 > \Delta = \tau^2 - 4\rho\lambda, \tag{56}$$

which is always true. $\qquad\square$

Proposition 4.11 appears to contradict Proposition 4.10. It does not since the case $m = 1$ where $A = a$ is a vector is non generic: in this case $\rho = [xx^T]$ is a scalar (and thus commutes with all matrices involved) and $B = (\varepsilon/\|a\|)aa^T$, and we can rewrite $N^T$ as

$$N^T = [(\rho - \frac{\varepsilon\tau}{\|a\|})(d \cdot a)a - \lambda d]a^T = -\lambda(\frac{1}{\|a\|^2}aa^T + \text{Id})da^T = -\lambda(\frac{1}{\|a\|^2}aa^T + \text{Id})aa^T. \tag{57}$$

In particular, since $((1/\|a\|^2)aa^T + \text{Id})$ is positive definite, $\text{Ker}(N^T) = \text{Ker}(aa^T) = \text{Ker}(ad^T)$, and we cannot conclude that $\mu$ is negative in the last part of the proof of Proposition 4.10. This is generically not the case for $m > 1$, so there is no contradiction.

