# OpenReview forum: "Dual Perspectives on Non-Contrastive Self-Supervised Learning"
_ICLR.cc/2026/Conference — ICLR 2026 Poster_

### Official Review · Reviewer_sCRT · 2025-10-26

**Soundness:** 3
**Presentation:** 3
**Contribution:** 3
**Rating:** 6
**Confidence:** 3

**Summary:**

The paper investigates two popular optimization approaches in non-contrastive learning: exponential moving average (EMA) and stop-grad (SG), both of which are employed in practice to prevent the model from feature collapse (that is, learning the trivial solution where model output is always zero). This question has been previously studied, starting from the work of Tian et al (2021), but the present work weakens some assumptions by relying on classic results from dynamical systems.

The main contributions of this paper are:
- to show EMA/SG does not optimize the original loss function and may not optimise any well defined function
- assuming the models are linear, the paper characterises the equilbria of the dynamical systems associated with EMA and SG, and proves that these equilibria are asymptotically stable

**Strengths:**

EMA and SG are often deployed in optimising deep learning models to prevent collapse. While the focus of this paper is on non-contrastive models (where these techniques are always used), there is a general need for understanding EMA/SG, and how these techniques can lead to desirable/undesirable solutions.

In particular, this paper provides theoretical/empirical evidence that EMA/SG do not minimise the original objective and the parameters need not necessarily converge, but the equilibria can be characterised as an (asymptotically stable) algebraic variety. These characterisations can provide a more informed understanding of when to (not) apply EMA/SG in deep learning or derive theoretically sound fixes.

**Weaknesses:**

1. The interesting results (Section 3) assume linear models: I note this first because it's a typical grudge about many theory papers. However, it's important to make two observations:
- The linearity is assumed with respect to trainable parameters. Notably, one could replace the linear non-contrastive models $f_\theta, f_\xi$ in Section 3 by random feature models or deep networks in lazy-training regime. This can be easily fixed in the paper by redefining $[xx^\top], [xy^\top]$. I believe this would not change any analysis. So, in principle, the analysis holds for models that are nonlinear with respect to input data.
- The linearity with respect to training parameters is necessary with our current theoretical understanding of EMA/SG. It is evident from the paper that interesting conclusions, which were not previously known, can still be derived assuming linearity. In summary, this "weakness" is just to explicitly refute any other possible review that criticises the linearity assumption in the paper

2. Vagueness of some proofs and assumptions. The presentation of some of the proofs is not formal enough (a few less-critical concerns are listed in questions). A bit more critical concern is the lack of clarity of loss in Section 3. Although not clearly stated, the loss is assumed to be a squared distance between the representations from the two models. Many of the proof steps rely on this fact. Here are a few observations:
- While the analysis of EMA/SG is motivated by papers on BYOL/SimSiam, both consider cosine-based loss functions and not exactly squared distance between (linear) model outputs. The normalisation plays a crucial role in these works, which cannot be explained by the current analysis.
- Some of the claims in Section 3 cannot extend to other losses. A simple example is the following case of SG: $\theta(0) = \xi(0) = 0; \dot{\theta}(t) = \cos(t); \dot{\xi}(t) = \theta(t) - \xi(t)$. In this case, WolframAlpha gives the solution: $\theta(t) = \sin(t)$ but $\xi(t) = (e^{-t} + \sin(t) - \cos(t))/2$. Hence, in this case, the continuous-time dynamics suggested in Equation (5) is an incorrect approximation to SG in discrete time. Similar examples can be constructed for EMA as well, raising doubt for the continuous-time dynamics.
- The proofs of Lemma 3.2 and Proposition 3.5 assume that the parameter $C$ follows $A$ (since it holds for SG in discrete time), but shouldn't these proofs include the dynamcis of $C$?

**Questions:**

The main questions are in point-2 of weakness. Several other minor comments are listed below:

1. Description of SG: I think the SimSiam paper eventually uses a symmetrized version of the loss, and hence, SG gets applied to both models. This is not considered in the present paper, where SG is applied only on $\xi$. It would be good to clarify this. Also line 99 (SG step-(a)) is confusing if one directly replaces $\xi_{t-1}$ by $\theta_{t-1}$ (because the update of second model is not clearly specified). The description of EMA with $\alpha_t=0$ is more clear.

2. lines 227-234: u,v are not defined and $\lambda$ is missing for inline expressions for $\nable \Omega$.

3. Proposition 3.5: Assumption $A$ is full rank is incorrect because it is a rectangular matrix. It is better to assume $S$ is strictly positive definite. By the way, is it possible to characterise when (if at all) is $S$ strictly positive at equilibria? This also may depend on the data, and may not hold generally.

4. Line 368: "Such a system admits $2^{m(m+1)/2}$ solutions."  It is not clear to me why this should hold. Please provide a reference or proof.

5. Theorem 3.8: Replace $A$ with a different notation (since $A$ is used in the model)

6. Lemma 3.4: The result that $B^\top B = AA^\top$ asymptotically is reminiscent of several results on gradient descent for deep linear networks (dating back to 1990s). It would be good to include a short discussion on this. Perhaps the asymptotic equality is a consequence of EMA/SG in this setting.

7. Line 665: "There is a priori no reason .." I don't believe this is a formal way to prove a statement. You need to provide a concrete counter-example. Consider a simple case of a data and loss function to argue that a critical point of $F$ may not be a critical point of $E$.

8. Line 687-688: Same issue as above. Use the linear case (discussed after the proof ends) and assume some simple fixed data to make your point. Perhaps assuming the augmentation is $y = x + \epsilon$ could work here.

---

> ### Author Response · Authors · 2025-11-20
> **Rebuttal response (1/2)**
>
> We thank the reviewer for their positive assessment and for clearly recognizing the motivation and significance of our contributions. We also appreciate the constructive suggestions, which helped us improve the clarity and rigor of the manuscript. In the revised version, we made substantive improvements to the proofs to make them precise and rigorous, while making assumptions explicit, as suggested. We address each point below.
>
>
> ### 1. **Connection between EMA and SG dynamics:**
> We acknowledge that the current presentation may make the link between discrete and continuous dynamics unclear. We will clarify this using the notion of two-time-scale dynamics: in EMA, the target network $\xi$ is updated on a faster time scale than the student network $\theta$. SG corresponds to the limiting case where the target moves so quickly relative to the student that $\theta$ is perceived  as effectively frozen by $\xi$; equivalently, $\xi$ fully tracks $\theta$ before $\theta$ changes. In the discrete-time EMA update, this effect can be mimicked by setting $\alpha = 0$: a single gradient step with step-size 1 exactly minimizes the quadratic $\xi \mapsto  1/2|\theta-\xi|^2$, yielding $\xi = \theta$.
> However, in continuous-time, the step-size is infinitesimal, so the target cannot instantaneously reach $\theta$. Therefore, one cannot recover SG dynamics by simply setting $\alpha$ in Eq. (5). Instead, SG should be understood as the infinite separation of time scales limit of EMA, where $\xi$ converges to $\theta$ arbitrarily fast. We will make sure to clarify this in the text.
>
> This should shed light on the following points:
>
> - **Why is the parameter $C$ assumed equal to $A$?** We should indeed clarify this point. In Eqs (4) and (5), we present the dynamics for EMA which involves a third variable C. The dynamics of SG involves only two variables (A and B). We believe the confusion might be come from sentences like “The dynamics for the SG algorithm are obtained by taking $C= A$ in Eq. (4) and not using the $C$ Update.” as if we derived the SG dynamics from the EMA one. We will clarify in the text that SG is described by two equations involving $A$ and $B$ variables only.
>
>
> - **The Wolfram Alpha example:** Thank you for constructing this example! The apparent contradiction is that $\xi(t)$ is not equal nor does it get closer to $\theta(t)$. However, there are two important points to take into account:
>
>     - **The exemple illustrates the EMA not SG dynmics.** Contrary to the discrete case, the dynamics in Eq. 5 does not recover SG dynamics by setting $\alpha=0$, as we discussed in the previous point. The correct continuous-time dynamics does not involve the variable $\xi$ (only $\phi$ and $\theta$). Hence, this example corresponds to the EMA dynamics, for which $\xi$ and $\theta$ might not be equal.
>
>      - **Why $\theta$ and $\xi$ do not get closer in this example?** Here the parameter $\theta$ never converges to a fixed point ($\theta*$), as such we cannot expect $xi$ to track $\theta$ accurately as the moving target keeps running away. On the other hand, whenever $\theta(t)$ converges to some value $\theta*$, it is easy to show that $\xi$ also converges to $\theta*$. The proof relies on the dominated convergence theorem after expressing $\xi$ explicitly in terms of $\theta$, i.e. :  $\xi(t) = \xi(0) + \int_0^t e^{-u}\theta(t-u) du. $ We will make sure to clarify that we are only interested in critical points and that $\xi$ does not need to closely approximate $\theta$ far from a critical point.
>
> ### 2. **Additional discussions/clarifications:**
>
> **Linear model setting:** Thank you, we will make sure to discuss possible extensions to the non-linear setting in the lazy/NTK regime.
>
>  **Clarifying the loss:** Indeed, while we define the loss in L228 of section 2.2, it can easily go unnoticed for the reader. We now recall that at the beginning of section 3.
>
> **Role of normalization:** We agree that normalization is often used in practice and helps with performance. Our analysis does not capture its effect and instead follows the setting from Tian et al. 2021. We will clarify this in the text.

---

> > ### Author Response · Authors · 2025-11-20
> > **Rebuttal response (2/2)**
> >
> > ### 3. **Answers to questions:**
> >
> > 1. **Description of SG:**
> >     -  We will clarify that we consider the dynamics without symmetrization to simplify the study which is the same setup as Tian et al. 2021.
> >     - Update equation in line 99: We hope that the discussion in “Connection between EMA and SG dynamics:” that we will add to the text will clarify this update.
> >
> > 2. **Missing notation:** We have now defined $u$ and $v$ in the text and added the missing parameter $\lambda$.
> >
> > 3. **Proposition 3.5:**
> >     - **Clarifying terminology** We now say 'maximal rank' instead of 'full rank' to indicate that $A$ is of rank $m$, the maximum possible rank that a matrix of dimensions $n \times m$ can achieve given that $m<n$.  We now use maximal rank instead of full rank.
> >
> >      - **Existence of a full rank solution.** Thank you for raising this important question. We have added a new proposition (4.7) where we establish the existence of a small perturbation to the original data distribution so that a maximal rank solution $A$ is guaranteed to exist. The argument relies on the implicit function theorem as well as a technical lemma (6.2) which establishes existence of solutions us Brouwer’s fixed point theorem. This shows that such solutions are generic: in the precise sense that: even though for a given distribution $P_0$ the property fails to hold, one can always slightly perturb it so that the property holds again.
> >
> > 4. **Number of solutions:** We clarify that these are based on a counting argument. This equation being symmetric, it is composed of $m(m+1)/2$ polynomial equations of order 2. Here, the variables are the $m(m+1)/2$ free coefficients of symmetric matrix $S$. According to Bezout’s theorem, the maximum number of (real and complex) solutions $p$ polynomial equations of degree $d$ is $d^{p}$. The result immediately follows. We will clarify this.
> >
> > 5. **Notation in Theorem 3.8**: We have replaced A by J, as suggested, to avoid confusion in the notation.
> > 6. **Lemma 3.4:** We now added a reference to early work from Baldi and Hornik 1989.
> >
> > 7. **Counter-example establishing critical points of $F$ are not necessarily critical points of $E$**. We now complement the proof, in the revised version, with a counter-example, as suggested, building on the linear setting to establish that one can always find a small perturbation of the original data distribution such that the desired property holds. The construction relies on the new proposition (4.7) that establishes the existence of non-degenerate critical points for the dynamics.
> >
> > 8. **Rigorous proof for inexistence of a potential:** We have now provided a complete proof constructing an explicit small perturbation to the data distribution for which the claimed property holds, when the encoder and predictor are networks with final linear layers (mild assumption).  The result relies on a technical result (Proposition 6.1) that shows the integrand in Eq. 16 cannot be identically 0. We then constructs a perturbation that puts more mass in a region where the integrand is non-vanishing, thus establishing the result.

---

> > > ### Comment · Reviewer_sCRT · 2025-11-25
> > > **As a reviewer, I apologise for the low quality reviews that this paper received**
> > >
> > > Thanks for your response. I have increased my score to 8, but I am not sure if we can have a meaningful discussion about this paper.
> > >
> > > It is becoming a common problem for theory papers in ICLR, NeurIPS, etc. that the reviewers do not even attempt to understand the math, and provide surface-level criticism about experimental validation, timeliness etc.
> > >
> > > I hope the AC takes note of the fact that there is a very limited rigorous understanding of the dynamics of non-contrastive learning, and we need to acknowledge works that provide fundamental mathematical results without immediate algorithmic benefits. Not providing decent reviews for such works has a long-term impact of demotivating theoreticians from working on such relevant topics.

---

> > > > ### Author Response · Authors · 2025-11-26
> > > >
> > > > Thank you very much for your thoughtful follow-up and for increasing your score.
> > > > We truly appreciate the time and care you put into understanding the theoretical contributions of our work.
> > > > We share your perspective on the importance of rigorous theoretical analysis for non-contrastive learning and we are grateful for your recognition of the value of such efforts.

---

### Official Review · Reviewer_f64b · 2025-10-27

**Soundness:** 2
**Presentation:** 2
**Contribution:** 2
**Rating:** 2
**Confidence:** 4

**Summary:**

This paper investigates the common self-supervised learning procedures of **stop gradient** and **exponential moving average**, which are used to prevent representation collapse.

The authors show that while these iterative procedures do not optimize the original or any other smooth objective function, they are effective at avoiding collapse.

Using a dynamical systems perspective in the linear case, the study demonstrates that minimizing the original objective *without* these procedures always leads to collapse, and it explicitly characterizes the **asymptotically stable equilibria** for both procedures as algebraic varieties, which are illustrated with empirical experiments.

**Strengths:**

1. Provides a clear theoretical foundation explaining why stop-gradient and EMA prevent representation collapse.
2. Offers rigorous analysis from both optimization and dynamical systems perspectives.
3. Empirical experiments effectively support the theoretical findings on real and synthetic data.

**Weaknesses:**

1. The statement in Lines 156–157 (“The SG and EMA training procedures have been designed to avoid collapse in self-supervised learning”) is not entirely accurate. Prior work, such as SimSiam, has already shown that Stop-Gradient alone can prevent collapse without EMA. Moreover, BYOL’s EMA inherently includes a Stop-Gradient operation, and the true collapse-prevention factor lies in the asymmetric *predictor* component — without it, SG or EMA alone would fail.

2. While the paper provides some theoretical analysis of non-contrastive learning, the discussed methods are now outdated compared to more recent and stronger approaches such as MoCov3, DINO, iBOT, and MAE-based self-supervised models. Several studies [1–3] have already offered deeper theoretical insights into both contrastive and non-contrastive frameworks, making the contribution of this paper less timely and impactful.

3. The experimental validation is weak, relying heavily on limited and synthetic data without robust quantitative evaluation on standard benchmarks. This weakens the credibility of the theoretical claims — more comprehensive and realistic experiments are needed to meet the expectations for publication.

[1] *How Does SimSiam Avoid Collapse Without Negative Samples? A Unified Understanding with Self-Supervised Contrastive Learning*, ICLR 2022

[2] *Understanding Collapse in Non-Contrastive Siamese Representation Learning*, ECCV 2022

[3] *Contrasting the Landscape of Contrastive and Non-Contrastive Learning*, AISTATS 2022

**Questions:**

See weaknesses

---

> ### Author Response · Authors · 2025-11-20
> **Rebuttal response**
>
> We thank the reviewer for their thoughtful assessment and for highlighting several strengths of our work, including the clear theoretical foundation, rigorous analysis, and supporting empirical evidence. We appreciate the constructive feedback and believe that the raised concerns largely reflect points of clarification or scope rather than substantive limitations. In particular, let us reassert that our goal is to provide a principled understanding of why stop-gradient and EMA (+ predictor)—techniques that remain central in state-of-the-art methods (e.g, Dino v2, v3)—effectively prevent collapse, rather than to benchmark a new architecture. We hope our responses below clarify these aspects and further underscore the relevance and scientific value of our contribution.
>
>
> 1. **Clarifying the statement in lines 156-157.** Thank you for pointing this out. We now clarify in the revised version that by “SG and EMA training procedures” we meant the whole dynamics that rely on a 1) predictor, 2) SG for the teacher and 3) optionally EMA for the teacher. Hence, no contraction with prior work.
>
> 2. **Why is the theoretical analysis timely?**
>     **A. SOTA work still rely on EMA:** The most recent and most performant SSL models ( Dino V2 (2024), V3 (August 2025)) rely on student/teacher networks with EMA. This is also the case of iBOT.
> The other models mentioned (MAE, MoCov3) are all largely outperformed by Dino V2/V3 (e.g. Imagenet-1k with Linear probing and ViT-L: 86.3% for DinoV2,  82.3% iBOT vs 81% for MoCoV3, 76.6% for MAE(ViT-H) (73.5 ViT-L)).
>
>     **B. The cited works do not provide formal studies for EMA or SG.**  The cited works are valuable contributions to the understanding of self-supervised learning. However, many important questions necessary for advancing scientific understanding remain open—some of which are addressed in our paper. Specifically:
>
>    - [1] ICLR 2022: The paper introduces conjectures on why stop-gradient may prevent collapse but does not prove them.
>    - [2] ECCV 2022: This is primarily an empirical study without theoretical guarantees explaining the underlying mechanism of collapse avoidance.
>    - [3] AISTATS 2022: This work analyzes an alternating optimization procedure where both networks optimize the same objective (e.g., Theorem 2 shows gradient-flow dynamics). This setup is fundamentally different from EMA or stop-gradient updates, which, as we formally show in Proposition 2.2, do not correspond to the gradient of any smooth objective and are thus, much more challenging to study.
>
> 3. **Experiments on SSL for action classification in videos in section 2.3.** We will emphasize in the revised version that these real data experiments illustrate a realistic setup for SSL, supporting the theoretical findings.

---

> ### Author Response · Authors · 2025-11-28
>
> Dear reviewer,
> we would really be grateful for you to take te time to reply to our detailed rebuttal. Thank you.

---

### Official Review · Reviewer_dqoh · 2025-10-28

**Soundness:** 3
**Presentation:** 2
**Contribution:** 3
**Rating:** 2
**Confidence:** 2

**Summary:**

In this paper, paper provides a theoretical study of two core techniques used in non-contrastive self-supervised learning (SSL) methods - stopping Gradient (SG) and Exponential moving Average (EMA) - that are essential to prevent representation collapse in algorithms such as BYOL and SimSiam. From a dual perspective of optimization theory and dynamic systems, the authors investigate why these methods have been empirically successful in the absence of negative samples and explicit regularization.

**Strengths:**

The topic of the paper is good. It tries to explain non-contrastive self-supervised methods like BYOL/SimSiam from both optimization theory and dynamical systems theory, which "learn good representations without seeming to have an objective function." This combination is innovative in the research of SSL theory. In particular, the perspective of dynamical systems (continuous-time analysis, equilibrium points, and stability proofs) provides new insights into the dynamics of self-supervised training. The authors verify that the non-convergence properties of SG and EMA are consistent with the theory on video classification tasks. Authors also conduct toy experiments to illustrate the trajectory of the dynamical system. Although the experiment is small-scale, it serves as an intuitive verification of the theoretical behavior.

**Weaknesses:**

1. The paper introduces several symbols on the same page, and these symbols represent different meanings in different contexts. This high density of symbolic definitions makes readers need to frequently backtrack and compare with the previous texts during reading, which increases the burden of understanding and is not conducive to quickly grasping the core deduction logic.
2. I understand the authors’ intention to introduce examples of SG and EMA directly in the introduction. However, such a structure can be risky if not carefully managed. The current presentation makes it challenging for readers to grasp the main content of the paper, the problem being addressed, and its practical significance.
3. The experiments only include two video classification tasks with low-dimensional toy simulations. There is a lack of comparison with other theoretical assumptions and a lack of quantitative metrics showing the extent to which the original objective function is not minimized.
4. In Proposition 2.1 and 2.2, "in general", "in practice", "it is unclear whether..." appear several times. The authors lack clarity on which cases are valid and which are exceptions.
5. Authors must not assume that all readers are familiar with technical terms, so authors need to approach the introduction of their work from the general ML audience perspective.

**Questions:**

1. In proving that P, Q is not a gradient field, the author uses the term "in general" several times, but does not explicitly state the specific conditions under which it applies. As a reader, it is difficult to judge whether there are exceptions to this conclusion. If there are such special cases, can you give an intuitive example or a theoretical explanation?
2. The analysis in Chapter 3 assumes that both encoder and predictor are linear maps, which leads to conclusions about the asymptotic stability of the dynamical system under this condition. But in real networks are usually nonlinear. Then, can this stability analysis obtained in the linear case be extended to nonlinear models?
3. Since α is assumed to be constant in the analysis, do the authors explore the impact of dynamic changes in α on stability or convergence?
4. After all the proofs and discoveries, how can this be implemented technically (like a solution or new method the authors propose based on the observation)?

---

> ### Author Response · Authors · 2025-11-20
> **Rebuttal response**
>
> We thank the reviewer for their positive and thoughtful assessment. We are glad that they found our theoretical perspective combining optimization and dynamical systems insightful and relevant, and we appreciate their constructive feedback on clarity and scope. We propose to carefully address the identified weaknesses through improved presentation, additional clarifications in the theory, and discussion of extensions beyond the linear setting.
>
> ## **Major points**:
>
> **Clarity:**  As we understood, points 1, 2, 4 and 5 are all about clarity and presentation. We thank you for raising these points and we agree that there is room for improvement. As suggested we have implemented the following improvements:
>
>   - **Simplifying notations:**
> We no longer introduce the notation $P$ and $Q$ for the vector fields and directly refer to them as $\nabla_{\theta} F$ and $\nabla_{\psi} F$. We believe the reading should now be easier as the main objects to keep track of now are only the losses $F$ and $E$ and their derivatives.
>   - **Clarifying the introduction:**
>     - To ensure the introduction is accessible to a wider audience and avoid  overloading it with technical terms and notations, we have moved the problem setting out of the introduction and to section 3.
>     - We also explicitly discuss how the work is positioned w.r.t. the literature, to clarify the importance of the question addressed.
>
>   - **Using precise terminology:** We apologize if some of the used terminology gave an impression of lack in clarity.
>     - In the revised version, we clarified what we mean by “in general”
> 	“Whenever we state  that some property holds in general, this means that, although they may not hold for certain data satisfying specific equations, they do hold for all generic data, in the standard mathematical sense, following the common notion of genericity in dynamical systems and algebraic geometry (e.g., Hirsch, Smale & Devaney, Differential Equations, Dynamical Systems & An Introduction to Chaos).”
>     - We could not find expressions like “in practice” and “it is unclear whether” in the statements of propositions 2.1 and 2.2. In the revised version, we made sure that any occurrences of those in the text are not about the results but only to motivate them: e.g. beginning of section 3.1.
>
>
> **Measuring lack of optimization in practice:** We tracked two metrics: the objectives whose partial gradients are computed, and the difference between successive iterates. The latter does not converge to 0 which is a strong indication that the algorithm does not converge and a fortiori does not minimize any objective. That being said, any suggestion is welcome.
>
> **Comparison with other theoretical assumptions:** We will be happy to consider other theoretical assumptions that are relevant to the problem considered. Please let us know if you have anything specific in mind.
>
>
> ## **Questions:**
>
>
> **Making the proof of absence of gradient field rigorous.** Thank you for encouraging us to improve the quality of the result. We now establish rigorously for a family of encoders and predictors that admit a linear last layer and are not identically vanishing the following: “given any data distribution, we can always construct an arbitrarily close distribution for which the corresponding vector fields are not gradients of any function.”
>
> **Analysis beyond the linear setting:** Thank you for pointing this out. We will clarify in the paper that there is a path towards a non-linear setting through the framework of Neural Tangent Kernels as pointed out by reviewer sCRT. This setting shares a lot of similarities with the linear dynamics, but can account for non-linear models. However, this would require: 1- a precise understanding of the dynamics in the linear setting (which is the aim of the current work) and 2- additional developments beyond the scope of this work.
>
> **Effect of dynamic change of $\alpha$:**
> We have kept the same setup as Tian et al. 2021, and leave such an investigation for future work. We agree it is an interesting topic.
>
>
> **Insights from the theory:** We clarify that this work does not pretend to give practical advice or introduce new algorithms. It is mainly concerned with providing a theoretical analysis supporting the use of EMA or SG.

---

> > ### Comment · Reviewer_dqoh · 2025-11-26
> > **Rebuttal response**
> >
> > Thank you for your reply.
> >
> > At the same time, I also reviewed the other responses provided by the author. I have raised my score.

---

> > > ### Author Response · Authors · 2025-11-26
> > >
> > > Thank you for your update and for the constructive feedback that helped us improve the paper.
> > > We appreciate the time and effort you dedicated to evaluating our work.

---

### Official Review · Reviewer_PrY7 · 2025-11-01

**Soundness:** 3
**Presentation:** 1
**Contribution:** 3
**Rating:** 4
**Confidence:** 3

**Summary:**

This work studies the role of two key components: stop gradient and Exponential Moving Average (EMA) for non-contrasitve from both optimization and dynamical systems perspectives. From the optimization perspective, they consider a general setting and prove that neither SG nor EMA can be interpreted as minimizing a well-defined smooth loss function, but they can avoid the representation collapse. From the dynamical systems perspective, they prove with SG and EMA that the dynamic system is asymptotically stable (assuming a linear setting). They also conduct real-world and simulation experiments to support their discussion.

**Strengths:**

* This work clearly shows the influence of SG and EMA from the theoretical perspective and conducts experiments to support their results.

**Weaknesses:**

* It would be better to provide a proof sketch for Proposition 2.2, which makes it easier for readers to understand why with SG and EMA, the $\bar{P}$ and $\bar{Q}$ are not the gradient fields of any smooth function.

* We know that the analysis for the nonlinear setting is hard. However, it would be better to discuss how to extend to a nonlinear NN (even a 2-layer Softmax NN).

**Questions:**

Question:

Q1: Why does the proof of this work require weaker assumptions compared with Tian et al?

Minor Comments: It would be better to use more formal writing, such as lines 247-248, and some experimental settings can be moved to the appendix. Then, there is more space for the proof sketch.

---

> ### Author Response · Authors · 2025-11-20
> **Rebuttal response**
>
> We thank the reviewer for their positive and thoughtful assessment of our work. We appreciate the constructive feedback and provide our responses to the raised points below.
>
> **Proof sketch for Proposition 2.2:**
> We agree that including a proof sketch would help readers grasp why SG and EMA are not gradient fields of any smooth function. We have added such a sketch in the main text, highlighting the key steps while keeping the full formal proof in the appendix for completeness.
>
>
> **Extension to nonlinear networks:**  Thank you for pointing this out. We will clarify in the paper that there is a path towards a non-linear setting, for instance, through the framework of Neural Tangent Kernels, as suggested by reviewer sCRT, which shares a lot of similarities with the linear dynamics, but can account for non-linear models.
>
> **On the weaker assumptions compared to Tian et al.**
> As noted by the reviewer, Tian et al. introduce additional assumptions in their proofs that are not required in our analysis. This difference likely arises from variations in analytical tools and derivation strategies. In our case, leveraging matrix identities allows for a simpler formulation that avoids these assumptions, as demonstrated in our proofs. We will clarify this distinction more explicitly in the revised manuscript.

---

> ### Author Response · Authors · 2025-11-28
>
> Dear reviewer,
> we would really be grateful for you to take te time to reply to our detailed rebuttal. Thank you.

---

### Author Response · Authors · 2025-11-20
**General response**

We thank the reviewers for their thorough feedback and for encouraging us to improve the quality and clarity of this manuscript. We have made the following major revisions in response to the core concerns raised:

**Rigorous and complete proofs:** We now provide complete proofs establishing that the claimed properties hold in a generic sense. By this we mean that given any data distribution, we can always find an arbitrary small perturbation to it so that the claimed properties hold. We do that for the propositions (Proposition 3.1 and 3.2 (previously prop 2.1 and 2.1)), and we have added a new proposition (prop 4.7) rigorously establishing the existence of maximal rank solutions to the EMA and SG dynamics in the linear setting. This result relies on Brouwer’s fixed point theorem and the implicit function theorem to establish the result in a generic sense, i.e. up to small perturbations of the data.

**Clarity of the introduction:** We have improved the introduction to ensure it remains accessible to a wider audience and avoid overloading it with technical terms and notations. We have moved the technical definitions out of the introduction into  section 3.

We believe these changes directly address the concerns raised while strengthening the overall contribution. Below, we provide detailed responses to each reviewer comment.

---

### Meta-Review · Area_Chair_gfon · 2025-12-19

**Summary:**

This paper presents a theoretical analysis of stop gradient (SG) and exponential moving average (EMA) training procedures in non-contrastive self-supervised learning. A key strength of the work is the rigorous theoretical grounding it establishes for these widely used heuristics, which previously lacked a clear optimization interpretation. Although the current analysis is primarily limited to linear setting (with discussed extensions to the neural tangent kernel regime), AC believes these results still constitute a meaningful and potentially foundational step for future research, warranting its inclusion in the conference. The recommendation is for acceptance.

**Reviewer Concerns:**

Many reviewer concerns have to do with the clarity of the presentation. The authors addressed these concerns to a significant degree by simplifying the dense notation and restructuring the flow of the paper for better readability. The authors also addressed the concern about contribution by providing rigorous proofs and clarified the distinction between their work and Tian et al. removing restrictive assumptions. The concern regarding the timeliness of the method was rebutted by noting that many SOTA models rely on EMA; while this empirical concern remains partially outstanding as the current analysis cannot be directly applied to SOTA models, AC believes this is an acceptable scope limitation similar to the linear model assumption.

**Reviewer Scores:**

Two of the reviewers (sCRT, dqoh) explicitly indicated that they would like to raise their scores above accepting threshold. Reviewer PrY7 would likely have raised their score after the authors provided the specific proof sketch and discussions that were requested. Reviewer f64b may not raise their score as the main critique is about the paper's empirical values.

---

### Decision · Program_Chairs · 2026-01-26

Accept (Poster)